# Reactive aldehyde chemistry explains the missing source of hydroxyl radicals

Xinping Yang[1,2,6], Haichao Wang [3,4,6], Keding Lu[1] ✉, Xuefei Ma[1], Zhaofeng Tan[1], Bo Long [5], Xiaorui Chen[1], Chunmeng Li[1], Tianyu Zhai[1], Yang Li[1], Kun Qu[1], Yu Xia[5], Yuqiong Zhang[5], Xin Li[1], Shiyi Chen[1], Huabin Dong[1], Limin Zeng[1] & Yuanhang Zhang[1] ✉

Hydroxyl radicals (OH) determine the tropospheric self-cleansing capacity, thus regulating air quality and climate. However, the state-of-the-art mechanisms still underestimate OH at low nitrogen oxide and high volatile organic compound regimes even considering the latest isoprene chemistry. Here we propose that the reactive aldehyde chemistry, especially the auto-xidation of carbonyl organic peroxy radicals ($R(CO)O_2$) derived from higher aldehydes, is a noteworthy OH regeneration mechanism that overwhelms the contribution of the isoprene autoxidation, the latter has been proved to largely contribute to the missing OH source under high isoprene condition. As diagnosed by the quantum chemical calculations, the $R(CO)O_2$ radicals undergo fast H-migration to produce unsaturated hydroperoxyl-carbonyls that generate OH through rapid photolysis. This chemistry could explain almost all unknown OH sources in areas rich in both natural and anthropogenic emissions in the warm seasons, and may increasingly impact the global self-cleansing capacity in a future low nitrogen oxide society under carbon neutrality scenarios.

Hydroxyl radicals (OH) are the most important oxidant in the atmosphere, which largely control the removal of most trace gas compounds from local to global scales, and determine the atmospheric self-cleansing capacity. The OH chemistry initiates the formation of secondary pollution (mainly haze and ozone)[1–4], worsens air quality, and threatens human health and the ecological environment. The OH radical also regulates the fate of greenhouse gases like methane, chlorofluorocarbon, ozone, and aerosols to affect the climate[5,6]. For example, tropospheric OH concentration decreased by $1.6 \pm 0.2\%$, largely contributing to the methane growth rate anomaly in 2020 relative to 2019[5]. Despite its critical role in atmospheric chemistry and

climate change, the budget of OH is still not well understood from local to global scales[7].

There is a long-standing puzzle that OH concentrations measured in environments with low nitrogen oxide (NO) and high volatile organic compounds (VOCs) are usually significantly underestimated by the state-of-the-art models[2,8–18]. Prior studies attempted to explain the deviation in the past decades and proposed numerous radical generation mechanisms. Leuven Isoprene Mechanism (LIM)[19,20] and self-reactions of peroxy radicals[10,21] were proposed mainly for the forested areas, whereas these mechanisms are still insufficient to explain the discrepancy between the OH observations and model

[1]State Key Joint Laboratory of Environmental Simulation and Pollution Control, State Environmental Protection Key Laboratory of Atmospheric Ozone Pollution Control, College of Environmental Sciences and Engineering, Peking University, Beijing 100871, China. [2]State Environmental Protection Key Laboratory of Vehicle Emission Control and Simulation, Vehicle Emission Control Center, Chinese Research Academy of Environmental Sciences, Beijing 100012, China. [3]School of Atmospheric Sciences, Sun Yat-sen University and Southern Marine Science and Engineering Guangdong Laboratory (Zhuhai), Zhuhai 519082, China. [4]Guangdong Provincial Observation and Research Station for Climate Environment and Air Quality Change in the Pearl River Estuary, Key Laboratory of Tropical Atmosphere-Ocean System, Ministry of Education, Zhuhai 519082, China. [5]College of Material Science and Engineering, Guizhou Minzu University, Guizhou, China. [6]These authors contributed equally: Xinping Yang, Haichao Wang. ✉e-mail: k.lu@pku.edu.cn; yhzhang@pku.edu.cn

results. Additionally, a parameterized mechanism in the form of $RO_2 + X \rightarrow HO_2$ and $HO_2 + X \rightarrow OH$ was proposed to reproduce OH radicals without $HO_2$ generation[11]. It is demonstrated that more detailed mechanisms are urgently needed to narrow the gaps of OH radical chemical mechanisms in the troposphere[10,11,19–21].

In this study, with a retrospective meta-analysis of the surface-based radical observation dataset spreading over the major city clusters in China (Supplementary Text 1, Supplementary Figs. 1–2, and Supplementary Tables 1–3) in warm seasons during 2006-2019[2,11,14–17,22,23], we showed the missing OH source is closely related to the oxygenated volatile organic compound (OVOC) chemistry. Based on the combination of quantum chemical calculations and observation-constrained model simulations, we report the detailed reaction processes of the autoxidation mechanism of high aldehydes and further confirm its importance by examination in numerous field studies worldwide.

## Results

### Characterization of the OH missing source

We revisit the OH budget in seven campaigns conducted in China in warm seasons and find that although the model (Methods, Supplementary Text 1, Supplementary Figs. 1–2, and Supplementary Tables 1–3) reproduces the observed OH at high NO conditions (>1 ppb), it underestimates the observed OH radicals at low NO conditions (<1 ppb), and the OH underestimation becomes systematically more severe with the decreasing NO concentrations (Fig. 1a and Supplementary Fig. 3). Possible measurement interference, derived from the decomposition of the Stable Criegee Intermediates or a new class of molecule (ROOOH), was proved to be negligible[15,17,22,24–29]. This confirms that OH underestimation by model simulation derives from the incomplete understanding of atmospheric radical chemistry. The experimental budget analysis was applied to cross-check the chemistry, by comparing the classical and well-known OH production rate and the total OH destruction rate (Methods and Supplementary Text 2). The OH production rate is the summed experimentally determined reaction rates of nitrous acid photolysis, ozone photolysis, ozonolysis of alkenes, and hydroxyl peroxy radicals ($HO_2$) plus NO as well as $O_3$. The total OH destruction rate is calculated by the product of the observed OH concentration and total OH reactivity ($k_{OH}$)[2,11,15–17]. It revealed a universal fact of unidentified missing OH sources for all

seven warm-season campaigns in China, especially around noontime and afternoon with low NO levels (Supplementary Fig. 4).

The correlation between missing OH sources and the inverse of NO reactivity ($1/k_{NO}$) is positive within low NO regimes NO concentration below 1 ppb, equivalent to $1/k_{NO}$ above 5 s (Supplementary Fig. 5a), indicative of the missing OH processes closely related to the level of NO. Here we find that adding VOC reactivity into the expression significantly improves the correlation between missing OH sources and $1/k_{NO}$ (Supplementary Fig. 5b). Among the VOC species, it is notable that missing OH sources show a perfect positive correlation with ratios of OVOC reactivity versus NO reactivity ($k_{OVOCs}/k_{NO}$), which is much better than the other sets of correlations by anthropogenic or biogenic VOCs (Fig. 1b). Coincidentally, high OVOC concentrations generally occurred during noontime and afternoon[30], consistent with the OH underestimation periods. Moreover, OVOCs play a crucial role in the total VOC reactivity, with a large fraction of about half (Supplementary Fig. 2). Thus, it indicated that OVOCs might be tightly related to the missing OH sources, and the missing OH sources from OVOC chemistry show a competitive relationship with NO.

### The reactive aldehyde chemistry mechanism

Quantum chemical calculations are used to explore the OH regeneration pathways related to OVOCs. Prior studies examined the H-migration rate constants of carbonyl organic peroxy ($R(CO)O_2$) radicals from C3-C5 aldehydes and found the H-migration rates dramatically increased as the carbon number increased[31]. For C3 and C4 aldehydes, the H-migration rate for peroxy radicals is predicted to be negligible compared to the reaction rate with NO at typical NO concentrations in the troposphere. For C5 aldehydes, the H-migration rate of $RC(O)O_2$ from iso-pentanal was estimated to be up to $0.58\,s^{-1}$[31], which will then overshoot the effect of NO to dominate the fate of peroxy radicals. Wang et al. further affirmed the high autoxidation potential of aldehydes under atmospheric conditions[32].

Here, we further determined the H-migration rate of the organic peroxy radicals derived from hexanals through the WMS[33]//M06-2X[34]/MG3S[35] method (Methods and Supplementary Text 3). The kinetic calculation results showed that the 1,7 H-migration with a rate constant of $0.321\,s^{-1}$ at 298 K is the most favorable pathway among all possible H-migration reactions based on the multi-structural transition state theory[36] with Eckart tunneling[37]

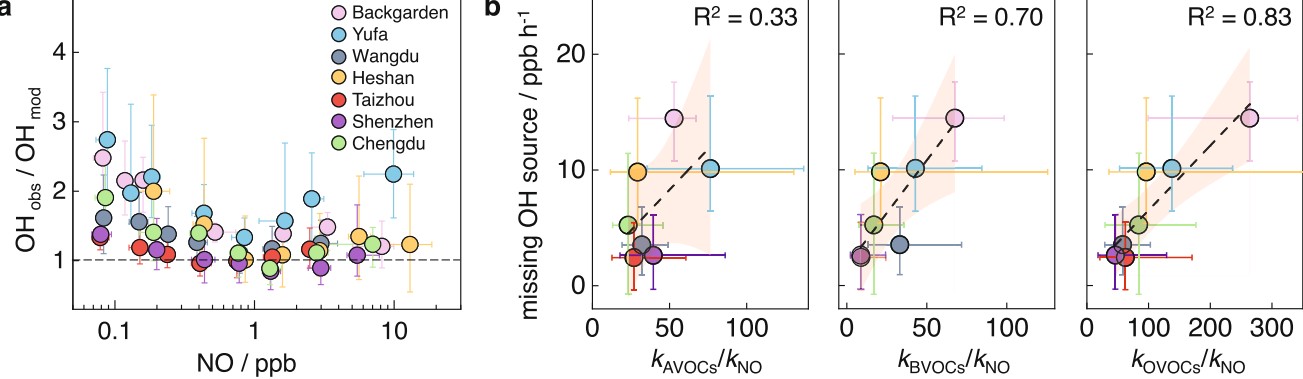

**Fig. 1 | The correlations between OH radicals and environmental parameters. a** NO dependence on ratios of observed to modeled OH ($OH_{obs}/OH_{mod}$) during daytime (around 08:00-17:00 local time) covering both high NO and low NO conditions in the warm-season campaigns in China. These field campaigns include: Backgarden, Guangdong in summer 2006[2,11]; Yufa, Beijing in summer 2006[14]; Wangdu, Hebei in summer 2014[15,73]; Heshan, Guangdong in autumn 2014[16]; Taizhou, Jiangsu in summer 2018[23]; Shenzhen, Guangdong in autumn 2018[22]; Chengdu, Sichuan in summer 2019[17]. The circles denote the median values, and the error bars denote the 25th to 75th percentiles. **b** Correlations of missing OH sources with the

matrix of OH reactivity, including ratios of anthropogenic volatile organic compound (AVOC) reactivity versus NO reactivity ($k_{AVOCs}/k_{NO}$), ratios of biogenic volatile organic compound (BVOC) reactivity versus NO reactivity ($k_{BVOCs}/k_{NO}$), and ratios of oxygenated volatile organic compound (OVOC) reactivity versus NO reactivity ($k_{OVOCs}/k_{NO}$) at low NO conditions (10:00-15:00 local time) in the seven warm-season campaigns in China. The circles denote the median values, and the error bars denote the 25th to 75th percentiles. The dotted line indicates the linear fitting curve. The shading shows the 95% confidence intervals.

(Supplementary Fig. 6a and Supplementary Table 4). It is noted that the conformers for reactants, transition states, and intermediate products are the lowest conformers, respectively. The molecules of reactant and transition states for generating conformers are shown in Supplementary Fig. 6b. Combining the estimations of both C5 and C6 aldehydes, we estimated the H-migration rate of organic peroxy radicals from higher aldehydes with the lower one. Thus, it represents a kind of conservative evaluation for this channel.

As shown in Fig. 2a, the R(CO)O$_2$ radicals undergo an H-migration process and are transformed to ·OOR(CO)OOH radicals, which then react with NO to produce ·OR(CO)OOH radicals. Subsequently, the ·OR(CO)OOH radicals mainly undergo two consecutive fast H-migration reactions and then react with O$_2$ to produce HO$_2$ radicals and hydroperoxyl-carbonyls (HPC)[31]. The photolysis of HPC is rapid, with a photolysis frequency as high as $10^{-4}$ s$^{-1}$ producing OH radicals, which has been proved by ref. 38. Here, for a conservative result, we set the HPC photolysis frequency to 10 times the methacrolein photolysis frequency, which is approximately $(0.6-0.9) \times 10^{-4}$ s$^{-1}$ around the noontime and corresponds to the lower limit of HPC photolysis frequencies[38]. The detailed reactions and rate constants of the higher aldehydes autoxidation chemistry are shown in Supplementary Table 5.

## Performance evaluation of the mechanism

We parameterized the Higher Aldehyde Mechanism (HAM) into the state-of-the-art atmospheric chemical mechanisms (RACM2, Regional Atmospheric Chemical Mechanism updated version 2)[39] and evaluated the contribution of HAM to radical chemistry (Methods). It is worth emphasizing that LIM, which involves the H-migration of RO$_2$ from isoprene[19,20], has been proven to be an OH regeneration mechanism that mainly works under high isoprene conditions[40]. Here, we set the box model with RACM2 as the base mechanism and further updated it

by incorporating LIM1 (LIM updated version 1)[15,19,20] and HAM. We diagnosed the averaged contribution of the unknown OH sources to the total OH production rates within the corresponding period (10:00-15:00) through the OH experimental budget analysis (Fig. 2b). It shows that the known OH radical sources are insufficient to match the total OH destruction rates. The unclassical OH sources accounted for 15-68% of the determined total OH production rates at low NO conditions in China.

Figure 2c shows the HAM explains 10–90% of unclassical OH sources in the seven campaigns. The contribution of HAM is always much larger compared to that of LIM1 for various campaigns, and the ratios of HAM contribution to LIM1 contribution vary between 1.7 and 8.2. At low $k_{OVOCs}/k_{NO}$ conditions represented by the Taizhou, Wangdu, and Shenzhen sites (Fig. 1), almost all unclassical OH sources could be explained within the uncertainties. With respect to the case of Backgarden, Yufa, Heshan, and Chengdu, unclassical OH sources were still missed by 68–86%, although the contribution of HAM is about 1–3 times larger than that of LIM1. The presence of still unexplained OH sources after including HAM in these campaigns may be tightly related to the following factors: (1) aldehydes from primary emissions, especially aldehydes with high carbon numbers (≥C8/C9) from biomass burning[41,42] or cooking activities[43,44], are not considered in the model, which are likely to be oxidized into RO$_2$ radicals with much higher H-migration rates than those currently used in this study; (2) the total concentrations of HPC groups were underestimated as they could also be produced from many other VOCs than higher aldehydes (e.g., long-chain alkanes[45], ethers[46], alcohols[47], alkenes[48]); (3) HPC photolysis rate for HPC species with different functional groups was underestimated. Additionally, aromatic autoxidation mechanisms have been proposed recently, demonstrating the significant role of OH-initiated oxidation of aromatics in forming highly oxidized products and thus secondary organic aerosols[49,50], while the mechanism was found negligible for OH

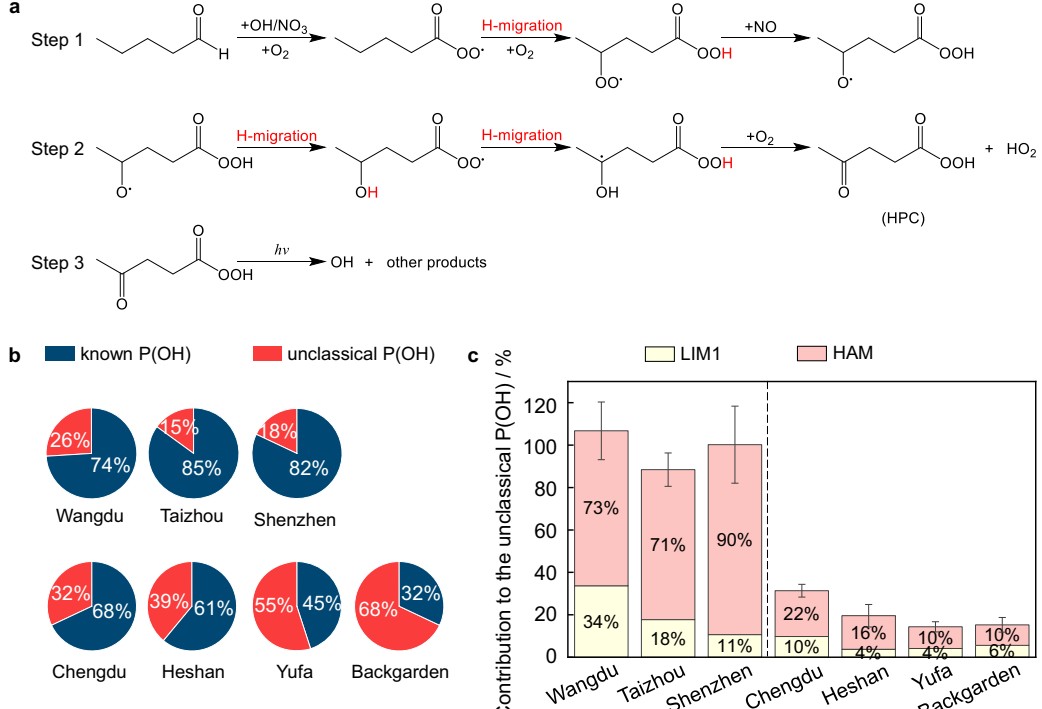

**Fig. 2 | The Higher Aldehyde Mechanism (HAM) and its contribution to OH regeneration. a** The detailed HAM processes that produce OH radicals, here take valeraldehyde as an example. Step 1 shows the chemical processes of ·OR(CO)OOH radical formation; Step 2 shows the two steps of H-migration to produce hydroperoxyl-carbonyls (HPC); Step 3 presents the HPC photolysis to produce OH radical. **b** The relative contributions of known and unclassical OH sources. **c** The relative contributions of the Leuven Isoprene Mechanism updated version 1 (LIM1) and HAM to the unclassical OH sources. The error bars denote the standard deviation.

generation in the seven field campaigns (Supplementary Text 4 and Supplementary Fig. 7). In short, the higher aldehydes play a much more profound role in the OH regeneration chemistry than isoprene, providing an important theoretical explanation for the unrecognized OH regenerations in China[11].

Recent studies have suggested that $RO_2$ derived from VOCs besides aldehydes, such as alkanes, aromatics, and other OVOCs (ketones, ethers, etc.), could also undergo autoxidation with possible subsequent generation of radicals[32,45,46,50–53]. It is reported that HPC-like substances are generated rapidly by two or more H-migration processes of $RO_2$ radicals[54]. Herein, we made a rule of thumb assumption, in which certain kinds of VOCs (including alkanes, alkenes, and aromatics, the detailed VOCs summed in Supplementary Text 5) could undergo OH oxidation process, successive H-migration processes and subsequently produce HPC-like reactive substances with a molar yield (named as Reactive Aldehyde Mechanism, RAM). RAM is a broad sense of HAM to some extent. We defined a simple index, named as φ, to characterize the generalized molar yield of OH regenerated by the oxidation of VOCs through the HPC chemistry. The index φ is derived from an inversed modeling method, to achieve a good agreement between the modeled and observed OH radicals (Supplementary Text 5). A reasonable φ of 0.4–2.2 is derived for our seven warm-season campaigns (Supplementary Table 6). Additionally, we show the incorporation of RAM into the model would further improve the agreement between the observed and simulated $HO_2$ concentrations to some extent (Supplementary Fig. 8).

The variation of the index φ in different campaigns may be caused by: (1) HPC species and HPC yield produced by different VOC oxidation processes may be highly varied[54]. (2) The OH yield from the HPC photolysis may be varied for different HPC species. (3) HPC photolysis frequencies which may differ by over four times[38].

From a global perspective, the atmospheric impact of RAM is evaluated through the OH recycling probability, which is a metric of the stability of atmospheric hydroxyl chemistry[55] (Methods). We focused on the seven warm-season campaigns in China and seven campaigns in other countries with reported radical observations and missing OH sources[9,10,12,27,56,57] (Supplementary Text 6 and Supplementary Table 7)—all together, fourteen campaigns across East Asia, Europe, and America (Fig. 3a). For each campaign, the OH recycling probability calculated by the observations and those simulated by the model sensitivity tests (Methods) were shown in Fig. 3b. The OH recycling probability observations were within 0.8–1.0 in most campaigns, without strong correlations with NO concentrations. The base model underestimated the OH recycling probability observations which denote the stability of the OH chemical system, especially under low NO regime, which is consistent with the missing OH regeneration processes in the base model. When the LIM1 and RAM mechanisms were added into the base model with a fitted φ (0.2–2.93), the discrepancy between the OH recycling probability observations and simulations was well eliminated (Supplementary Table 6).

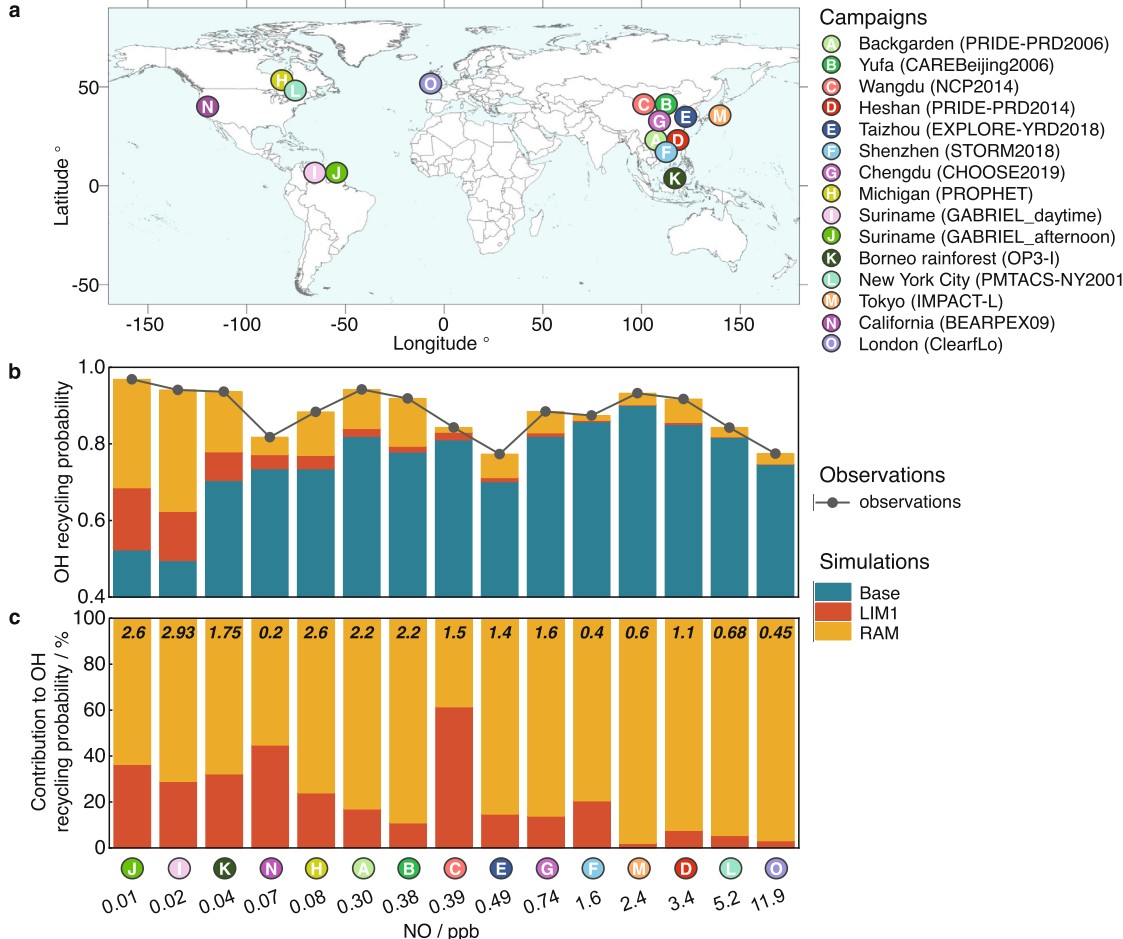

**Fig. 3 | Atmospheric impact of the reactive aldehyde chemistry. a** Location of the fourteen global campaigns. The basemap is from the Resource and Environment Science and Data Center, Chinese Academy of Sciences (https://www.resdc.cn/data.aspx?DATAID=205). **b** The observed and modeled OH recycling probability under different NO concentrations. The modeled results are from three scanerios, namely Base (well-known and classical OH formation mechanisms), LIM1 (Leuven Isoprene Mechanism updated version 1), and RAM (Reactive Aldehyde Mechanism). **c** The relative contributions of LIM1 and RAM to OH recycling probability under different NO concentrations, the number labeled at the top shows the fitted index φ in the corresponding campaign.

We subdivide the NO concentration intervals into ultra-low NO regime (NO < 0.1 ppb, mainly forested areas), low NO regime (0.1 ppb < NO < 1 ppb, mainly suburban areas), high NO regime (1 ppb < NO < 10 ppb, mainly urban areas), and ultra-high NO regime (NO > 10 ppb, mainly urban center of megacities). The importance of LIM1 and RAM mechanisms varies greatly at different NO intervals (Fig. 3c). The impacts of LIM1 on the OH recycling probability were significant under the ultra-low NO regime (forested areas). Nevertheless, the RAM mechanism acts over a wide range of NO concentrations, covering both suburban and urban areas (corresponding to low and high NO regimes). The contributions of autoxidation mechanisms involving LIM1 and RAM to OH recycling probability are insignificant in ultra-high NO regimes (megacities) where the underestimation of OH recycling probability is non-severe.

### Increasing importance in the future

A conceptual model is proposed to highlight the role of RAM on OH regeneration in different geophysical areas (Fig. 4). Under the forested areas, the fresh plume is characterized by ultra-low NOx concentrations and high BVOC reactivities, and thus the OH productions belong to the LIM1-dominated areas besides the classical mechanism. While with the air masses aging, the primary BVOCs may be gradually oxidized to OVOCs and the LIM1-dominated areas are gradually transformed into RAM-dominated areas. Under the suburban and urban areas with a NO concentration range of about 0.1–10 ppb, the OH generation rates from RAM are proved to be fast to compete with their reactions with NO. Additionally, for the urban plume with high AVOC concentrations, OVOC concentrations continue to increase over the photochemical development due to the AVOC oxidation, thereby the aging air masses are more conducive to RAM. Under ultra-high NO concentrations, although carbonyl $RO_2$ autoxidation remains effective[32], the RAM cannot compete with the bimolecular reactions of $RO_2$ with NO anymore, so the effects of LIM1 and RAM on OH production become negligible (Supplementary Text 7 and Supplementary Fig. 9). With the implementation of global pollutant control policies, especially the proposal of a 'carbon neutral' strategy goal in China and the global, NO concentrations in megacities will be greatly reduced to the high or low regime, or even ultra-low NO regime. Under this condition, the contribution of the RAM mechanism to OH regeneration and OH recycling would get significant. Thus, we should be alert to the potential effects of these autoxidation mechanisms on tropospheric oxidation capacity.

## Discussion

In summary, our study unfolds a noteworthy mechanism of the OH regeneration pathway which was previously overlooked in the ambient conditions. The finding is contrary to the conventional view of OVOCs

as a sink of OH radicals and provides new insights into the long-standing puzzle regarding missing OH sources in the past decades. The higher aldehyde autoxidation mechanism enriches the knowledge of OH regeneration chemistry discovered on top of the isoprene autoxidation chemistry. Compared to the LIM1, the RAM does not only work in areas rich in natural emissions but also contributes significantly in areas rich in anthropogenic emissions, demonstrating the importance of RAM in sustaining high OH levels at broader environmental types globally. Moreover, the chemical mechanisms involving OVOC species deserve to be further reformed due to their high concentration and reactivity and the lack of attention in the mainstream chemical mechanisms (e.g., RACM2, Master Chemical Mechanism). For example, the kinetic parameters of $RO_2$ autoxidation from high aldehydes, involving rate constants and product yields, and the photolysis process of hydroperoxyl-carbonyls need to be further quantified through both quantum chemical calculations and kinetic experiments. This would be critical for air pollution control and climate change mitigations, especially in the incoming low NOx society in the future.

## Methods
### Radical observations

The observations of radical concentrations in China during 2006–2019 come from our seven warm-season campaigns, including CareBeijing2006 (Yufa site) and NCP2014 (Wangdu site) campaigns conducted in North China Plain; PRIDE-PRD2006 (Backgarden site), PRIDE-PRD2014 (Heshan site), and STORM2018 (Shenzhen site) campaigns conducted in Pearl River Delta; EXPLORE-YRD2018 (Taizhou site) campaign conducted in Yangtze River Delta; CHOOSE2019 (Chengdu site) campaign conducted in Szechwan Basin (Supplementary Fig. 1). These sites located in the four major urban agglomerations in China, all suffering from severe secondary pollution such as both high $O_3$ and $PM_{2.5}$ concentrations. During these campaigns, radical concentrations are measured by the laser-induced fluorescence system (LIF) based on the fluorescence assay by gas expansion technique[2,18]. Both OH and $HO_2$ concentrations were measured in the seven field campaigns, and $RO_2$ measurements were only carried out at the Wangdu, Heshan, and Chengdu sites. Besides radical concentrations, a comprehensive set of meteorological and chemical parameters was measured, including the temperature, pressure, relative humidity, photolysis frequency, OH reactivity, and the concentrations of related trace gases (NO, $NO_2$, $O_3$, VOCs, etc.), as shown in Supplementary Text 1 and Supplementary Tables 1–3.

### Radical experimental budget analysis

The numerical radical closure experiment constrained by the observations is applied to examine the state-of-the-art chemical mechanism

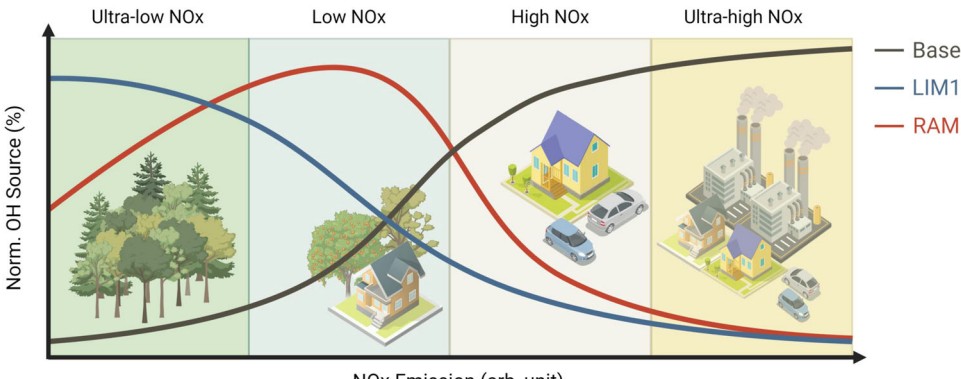

**Fig. 4 | Conceptual model of the role of reactive aldehyde chemistry in OH formation.** The OH formation is dominated by different mechanisms that largely depend on the NOx emission intensity. The Base is the well-known and classical OH

formation mechanisms, the LIM1 is Leuven Isoprene Mechanism updated version 1, and the RAM is the proposed Reactive aldehyde Mechanism (Created with BioRender.com).

by use of the short lifetime of the HOx radicals[3,58]. In this study, we mainly conducted the OH experimental budget by comparing the OH production and destruction rates. As introduced in detail in Supplementary Text 2, the OH production rate is the sum of primary sources (HONO photolysis, $O_3$ photolysis, ozonolysis of alkenes) and secondary sources (dominated by $HO_2 + NO$ and $HO_2 + O_3$). The total OH destruction rate is directly obtained from the product of the observed OH concentration and the observed $k_{OH}$ values. The discrepancy between the OH destruction and production rates denotes the missing OH sources.

### Quantum chemical calculations and kinetic methods

The reactants, transition states, and products were optimized using CCSD(T)-F12a/cc-pVDZ-F12[59] for the H-migration of carbonyl peroxyl radical ($CH_3C(O)O_2$). Benchmark values are calculated using W3X-L[60]//CCSD(T)-F12a/cc-pVDZ-F12 for $CH_3C(O)O_2$. Here, the WMS[33]//M06-2X[34]/MG3S[35] method was considered to be the most accurate method by comparing the results between different methods and W3X-L//CCSD(T)-F12a/cc-pVDZ-F12 method (Supplementary Text 3, Supplementary Fig. 10, and Supplementary Table 8). Thus, we used WMS//M06-2X/MG3S method to investigate the H-migration process of $R(CO)O_2$ radicals derived from hexanes (Supplementary Fig. 6a and Supplementary Tables 9–12). Intrinsic reaction coordinate (IRC)[61–63] were also calculated to show that the transition state connects with the corresponding reactant and intermediate product (Supplementary Figs. 11–15). Reaction rate constants of H-migration were calculated by using the multi-structural transition state theory[36] with Eckart tunneling[37]. Additionally, the daytime temperatures of the seven campaigns were all above 295 K (Supplementary Fig. 16), indicating that temperatures were not so low that it would overestimate tunneling probability. The electronic structure calculations were executed using the Gaussian 16[64], Molpro 2021[65], and MRCC computer codes [www.mrcc.hu.][66,67]. and the rate constants were calculated using KiSThelP software package[68] and $F_{fwd}^{MS-T}$ were calculated using MSTor 2017 program package[69,70].

### Framework for sensitivity test of HAM and LIM1

The radical simulation was carried out based on a zero-dimensional box model constrained with the observed photolysis frequency, meteorological parameters, and the critical trace gas compounds in five-minute averaged values. The box model is running with a spin-up time of two days and the dry deposition rates were set according to that reported in Wesely et al. [71]. The uncertainty of the model simulation is a combination of uncertainties in the measurements used as model constraints and reaction rate constants, and it is approximately 40%[2,15]. Here, we used the RACM2 as the base mechanism and conducted sensitivity experiments by adding the LIM1 (Sensitivity test 1) and HAM (Sensitivity test 2).

**Sensitivity test 1**. The LIM1 was proposed by ref. 20 and updated by ref. 19. In this study, we incorporated the updated LIM1 in the base model to evaluate the contribution of LIM1 to the OH production rates[15].

**Sensitivity test 2**. Based on our quantum chemical calculations about the H-migration of $R(CO)O_2$ radicals from higher aldehydes, the reported generation of HPC-like substances by H-migration processes[54], and the reported photolysis of HPC[38], we summarized the chemical reactions and reaction rate constants of the HAM in Supplementary Table 5. The H-migration rate constants of $R(CO)O_2$ radicals from higher (carbon number larger than four) aldehydes are estimated to be fast enough to compete with or even dominate over their bimolecular reactions with NO in the troposphere. The higher aldehydes dominated the ALD (propanal, butanal, and larger R-groups aldehydes) concentrations, with a contribution of 62%–81%

(Supplementary Figs. 17–18). Moreover, the H-migration rate constants of $R(CO)O_2$ radicals increase with increasing carbon numbers for R-groups. Here, we used the H-migration rate constant of $R(CO)O_2$ radicals from hexanals to denote that of higher aldehydes.

### Calculation of OH recycling probability

The stability of OH chemical system mainly could be affected by emissions of NOx, CO, and $CH_4$. The OH recycling would be inefficient under low NOx levels. When CO and $CH_4$ are very high and growing rapidly, such conditions can become catastrophic as both $O_3$ and HOx are removed. Conversely, when NOx concentrations are high, the OH recycling is efficient and the system could become autocatalytic, causing a runaway of oxidants. Such high-NOx conditions would lead to efficient OH recycling on the one hand, and on the other hand, it would lead to an increase in OH removal due to the reaction of OH and $NO_2$. Herein, to evaluate the effect of HAM on radical chemistry, we utilized the OH recycling probability, calculated from primary OH formation and OH recycling, to reflect the stability of tropospheric hydroxyl chemistry.

OH recycling probability ($\gamma$) was once defined by Lelieveld et al. in which both OH primary production rate and secondary production rate are taken into account[55]. The oxidation power ($G$) denotes the gross OH formation, which is the sum of primary ($P$) and total secondary ($S$) OH formation, as shown in Eq. (1).

$$G = P + S = P + \gamma * P + \gamma^2 * P + \ldots = \frac{P}{1 - \gamma} \tag{1}$$

Thus, the OH recycling probability is calculated by Eq. (2).

$$\gamma = 1 - \frac{P}{G} \tag{2}$$

When $\gamma$ approaches 1, it is indicated that OH radicals are insensitive against primary pathways and are almost entirely from secondary generation pathways. Under such conditions, the OH recycling is very efficient and the OH chemistry system could become autocatalytic. The $\gamma$ being approximately equal to 0.5 denotes the OH formation is quite efficient but not autocatalytic, demonstrating the OH chemical system may be stable. Under low $\gamma$ conditions, OH radicals are mainly from primary pathways, and thus, greatly sensitive to $O_3$ photolysis, etc. Overall, for the evaluation of the aldehyde chemistry, OH recycling probability is a comprehensive indicator from the perspective of primary and secondary generation pathways.

## Data availability

All data are available from the corresponding author upon request.

## Code availability

The main figures are produced by the IDL (Interactive Data Language version 8.3). The code for the model simulation can be obtained from the figshare repository under accession code https://doi.org/10.6084/m9.figshare.24993942[72].

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

## Acknowledgements

K.L. received financial support from the National Natural Science Foundation of China (grants 22325601, 22221004). K.L. received financial support from the National Key Research and Development Program of China (grant 2023YFC3706100). X.Y. received financial support from the National Natural Science Foundation of China (grant 42205111). H.W. received financial support from the National Natural Science Foundation of China (grant 42175111). H.W. received financial support from the National Key Research and Development Program of China (grant 2023YFC3710900). H.W. received financial support from the Fundamental Research Funds for the Central Universities, Sun Yat-sen University (grant 23lgbj002).

## Author contributions

K.D.L. and Y.H.Z. conceived the study. X.P.Y., H.C.W., and K.D.L. analyzed the data and wrote the manuscript with inputs from Y.H.Z. B.L., Y.X., Y.Q.Z. conducted the kinetic parameters from ab initio calculations. X.F.M., Z.F.T., X.R.C., C.M.L., T.Y.Z., Y. L., K.Q., X.L., S.Y.C., H.B.D., L.M.Z. helped to collect the field observation data. All authors contributed to the discussed results and commented on the manuscript.

## Competing interests

The authors declare no competing interests.
