## [Peer Review File · Nature Communications]

Reactive Aldehyde Chemistry Explains the Missing Source of Hydroxyl RadicalsReviewer #1 (Remarks to the Author):

This work addresses the issue of the underestimation of OH concentrations by theoretical atmospheric models in the low NO and high VOC regime.

Although my own work lies within the atmospheric chemistry field, I was actually unaware of this very interesting problem. Because of this, I cannot give a full account on how much effort has been put over the years into clarifying this divergence between experimental observation and theoretical modeling. However, the authors did a great job in contextualizing and explaining the scientific problem, revealing that the scientific community still has an incomplete understanding (at least until now, in this reviewer's perspective) of the chemistry behind OH regeneration.

The authors tackle this problem by performing a retrospective meta-analysis of a comprehensive warm-season field observation dataset and also by performing a theoretical study, discovering that the autoxidation of carbonyl RC(O)O_2 radicals derived from higher aldehydes could prove to be a novel and noteworthy (again, in this reviewer's eyes) OH regeneration mechanism that could explain the missing source of OH radicals.

As a theoretical and computational chemist I prefer not to make comments about the experimental observations. However, I can provide some comments on the theoretical calculations.

The authors start with hexanal, a C6 aldehyde, which suffers oxidation mainly by OH and then reacts with O_2 to yield the respective R(CO)O_2 radical. The latter radical then undergoes an H-migration reaction, for which its rate constant is calculated with multi-structural transition state theory (MS-TST). The choice of MS-TST is completely adequate for this problem, as also the level of theory used for the respective electronic structure calculations that support this rate constant, based on their careful benchmark calculations. However, the Supplementary Information (SI) does not provide enough information to make a full analysis or reproduction of the calculations. I believe that, at this stage, this manuscript lacks technical information in order for it to be more consistent and for it to better support the conclusions made by the authors. The authors may consider the following suggestions/questions:

- The total number of calculated conformers (reactants and transition states, TSs) and their relative energies (in a separate Table) should be mentioned somewhere in the "Text S3" part of the SI. Figure S6 only shows one reactant R(CO)O_2 conformer, but there should be much more, given the number of rotatable bonds in this molecule.
- Do the five shown TSs all connect to that one specific reactant conformer as shown in Figure S6?
- Are there only five TSs or these are only the lowest energy ones?
- Please provide Cartesian coordinates for all calculated reactant and TS conformers.
- Please provide the value for all imaginary frequencies of the TSs, along with the respective energetic data of the relevant stationary points that are needed to calculate the Eckart tunneling correction.
- Do the dashed lines in Figure S6 mean that there were Intrinsic Reaction Coordinate (IRC) calculations performed? If so, please state this in "Text S3". If not, how do the authors know that those connections in Figure S6 exist?
- Could the authors specifically state the type of H-migration (1,x) associated to each TS, for clarity purposes?
- The Eckart tunneling correction is well known to be overestimated at low temperatures (clear and very recent examples are shown here: DOI: 10.1039/d2ea00164k), so it is possible that some of the values shown in Table S5 of the present manuscript are overestimated. Such a possibility should be mentioned in the "Text S3" part of the SI.
- The previous point leads into this next one: how exactly were the RACM2 simulations performed? If the reader, like me, does not know anything about this simulation procedure, then some more details should be given. For example, at what temperature was the simulation performed? Or was it performed at several temperatures? This is important because of the concerns raised in the previous point, which could undermine some of the conclusions of this paper. I assume, however, that because the manuscript refers to "warm-season" several times, the simulation(s) was performed at 298 K, a temperature at which the overestimation of the Eckart tunneling correction is usually not troublesome. If so, please confirm this.

- Please cite the original RACM2 paper in the main manuscript,
<http://dx.doi.org/10.1016/j.atmosenv.2012.11.038>

The comments made above are mainly related to the first H-migration reaction of Figure 2A in the main manuscript. The authors then support the remaining reactions of Figure 2A with references. Specifically, in Table S6, they claim that "The rate constants of Nos. 3-5 were from Wang et al.", which for what I could understand is reference 25 of the SI:

<https://pubs.acs.org/doi/10.1021/acs.est.7b02374> However, I could not find the rate constant values for reactions 3-5 of Table S6 in this reference 25 paper. Could the authors provide a clear explanation of how they came up with these numbers? This is very important in order to make a good case for their conclusions.

Reviewer #2 (Remarks to the Author):

Hydroxyl Radicals (OH) is the most important oxidant in the atmosphere, it is critical to understand the atmospheric chemistry as well as the climate, since it determines the fate of almost all air pollutants and reactive greenhouse gases. This study aims to solve a very critical and fundamental problem in the field of atmospheric free radical chemistry, that is, the systematic underestimation of OH radical concentration by existing models under low NO and high VOC regimes. They used the most comprehensive field measurement datasets of atmospheric radical and precursors in the world, as far as I know, to identify the missing source of tropospheric OH radicals, they revealed the higher aldehydes, a typical class of oxygen volatile organic compounds that less concerned in the atmospheric chemistry, play an important role in the regeneration of OH. The new regeneration of OH mechanism by reactive aldehydes is clarified by a series of quantum chemical calculations and well assessed by the field datasets. In the end, they also argued that this new chemistry mechanism proposed in the study would be more important in the low NO_x conditions in the future.

The methods of data acquisition appear sound to me, conclusions are fully supported by data and well-argued based on relevant literature. I'm convinced and impressed by discussion data from different field campaigns across the world to draw the conclusions, which nicely emphasizes the locally wide spread importance of the findings. Overall, this study is well designed and nicely presented. This novel work is potentially important for understanding the atmospheric radical chemistry, and the article will attract widespread attention and have a far-reaching impact in the community of atmospheric and climates sciences, and beyond. Thus, I would like to recommend this interesting article for publication after addressing the minor comments shown below.

1. The first part of the Main Text seems a little confused. The background is introduced and followed by the meta-analysis of the field OH budget. I suggest the author adding a section header to distinguish the background (or Introduction) from the analysis of OH missing source. And in the end of the introduction section, a brief introduction about the method and purpose of this study should be outlined.
2. This paper proposed the OH regeneration mechanism by the quantum chemical calculations method, we understand that the fine quantification of these key kinetic parameters of higher aldehydes chemistry is a very long-term and large project, it would be too harsh to ask for achieving it in this study. But a clearer outlook should be provided to direct the laboratory studies in the future. I would like to see the author add some discussions in the last paragraph.
3. Line203-210, the explanation of high HPC yield seems to be plausible, but related references should be added to support this argument, especially the cooking emission source of high aldehydes. I suggest the author add the temperature information into the Table S7 to clarify the relationship of temperature and the HPC yield.
4. Line 165-174, why the HAM can largely explain the missing source but still cannot well address this issue in Backgarden, Yufa, Heshan, and Chengdu? Are there air mass conditions differences or something else, more discussions should be added to address this point.

Reviewer #3 (Remarks to the Author):

Review on "Reactive Aldehyde Chemistry Explains the Missing Source of Hydroxyl Radicals" by Yang and Wang et al.

The authors propose that higher aldehydes represent a significant OH source in the atmosphere, under relatively low nitrogen oxide concentrations. They show that the ratio between the observed and modeled OH concentration exceeds unity at low NO concentrations in seven measurement locations within China that serve as the key observational data set for this study. The authors further suggest that the unexplained OH production rate correlates with the ratio between NO and OVOC concentration. Based on a previous quantum chemical calculation study, they formulate a hypothesis that atmospheric aldehydes, comprising five or more carbon atoms, undergo autoxidation yielding a hydroperoxyl-carbonyl (HPC) and an HO₂ radical. They note, based on one another theoretical study, that HPC can undergo rapid photolysis yielding an OH radical. The authors perform estimations on the H-migration speeds for higher aldehydes (definition is unclear) using 1,7 H-migration speeds from quantum chemical calculations performed for R(CO)O₂ radicals derived from hexanal between 200 and 340 K ($\Delta T = 20$ K). They further parameterized the higher aldehydes autoxidation mechanism (termed HAM) into the RACM2 model and evaluated the contribution of HAM to the observed radical chemistry. The base case set for RACM2 included the well-established OH production rates from HONO and ozone photolysis, alkene ozonolysis, and the HO₂+NO and HO₂+O₃ reactions. In addition, it includes H-migration from isoprene-derived RO₂s (LIM).

The classical OH production channels could explain 15–68% of the OH production (POH) across the seven sites. The importance of the HAM in explaining the remaining OH production seemed to be highly pronounced for the measurement sites, where the classical OH production channels already explained the majority of the OH. In contrast, the modelled importance of HAM was lowest in the sites where OH production could not be reproduced with the classical production channels. HAM was reported as more critical than LIM in OH production across the sites.

Next, the authors suggest (with support from a number of references) that certain VOC species can produce HPC-like reactive aldehydes via autoxidation. As the molar yields for such aldehydes remain unresolved, the authors derive the required HPC yields to bridge the gap between the modeled and measured OH concentrations through sensitivity studies. The molar yields range from 0.1 to 35 for the different sites, and the highest yields correspond to the measurement locations, where HAM showed little relevance in OH production in the previous RACM2 simulations, and the unexplained OH production fraction was the highest. The authors suggest that the need for the very high HPC yield for these measurement sites could arise from temperature effects on HPC formation as the two sites experienced high temperatures albeit based on the shown data in the SI, the temperature differences were not significant.

The new mechanism including the specified VOC autoxidation to HPC with the determined HPC yields is then included in RACM2 (RAM simulations). Because of the fixed yields, the model evaluation could not be performed via OH concentration comparisons, as the "fitted" HPC yields would guarantee a perfect agreement. Instead, the authors evaluate the modeled and observed OH recycling probabilities and state that a discrepancy that exists in the RACSM2 base case is now eliminated. The authors finally suggest that the HAM mechanism is important over a wide range of NO concentrations and could be highly important in the future due to emission regulations yet the LIM mechanism dominates in the cleanest environments.

Overall, I do find the manuscript interesting and also interesting to a broader audience as understanding the OH budget is crucial from both climate and air quality perspectives from multiple angles. The proposed idea is very good and exciting, and the autoxidation of aldehydes has also been discussed previously in experimental contexts. However, I am unfortunately not convinced about the presented proof of the hypothesis, the quality of the analysis, and the descriptions of the utilized methodologies. Therefore, I cannot recommend the publication of this paper in its current state in Nature Communications.

I have listed a few suggestions below on how to improve this manuscript:

1. Could you perhaps evaluate how reasonable the utilized HPC molar yields are? The average

temperatures do not look that different in Table S3. Yufa has a lower temperature than Wangdu, but the HPC yield for Yufa is almost two orders of magnitude higher. Are the HPC yields used as constants per measurement sites as shown in Table S7?

2. Could explain the methods behind the observed OH recycling probability? If this is used as some sort of validation metric for your proposed hypothesis, I believe it is crucial to explain it thoroughly.

3. How does the measured HO₂ concentration compare with your RAM simulation outputs? Are there other ways you could evaluate your model results? How can you be sure that what you propose is actually happening?

4. Regarding the aromatic autoxidation mechanism discussed in L174–L177, you might be interested in the very recent publication by Iyer et al. (2023) and the discussion related to the findings reported by Wang et al. (2017).

5. The discussion about aldehyde autoxidation is also presented in Wang et al. (2021) could perhaps interest the authors of this manuscript.

6. I would also be curious to hear your motivation for the discussion in L262–L264 presented in your manuscript on the effects of NO on RAM. In the Wang et al. (2021) study, they measure increases in the molar yields of alkane-derived oxygenated products (incl. RO₂) as a function of NO. They suggest that multi-step isomerization of RO and/or RO₂ radicals are likely for all types of alkanes even at high NO concentrations (they went up to roughly 10 ppb). How would accounting for such change your results and the OH_{obs}/OH_{mod} under high NO concentrations?

7. The presentation of field data should be improved as they motivate much of the modeling work.

a. I would suggest presenting Figure 1A for all measurement sites separately in the SI without any NO-dependent binning.

b. The distributions of the observed NO concentrations and OH_{obs}/OH_{mod} could be useful. These plots could help with justifying such binning shown in Fig. 1A – a description that remains missing.

c. I strongly encourage you to show the range of variability in OH_{obs}/OH_{mod} per each NO bin e.g., with error bars.

d. It was also hard to notice at first glance that the x-axis in Fig. 1A is logarithmic. Could you perhaps add more tick marks to the axis to clarify this, or write it in the caption?

e. You mention in the SI that the Taizhou data points should be interpreted with caution because kOH was not measured. Could it explain why the trend is not so clear for the lowest NO concentration for that given campaign?

f. Fig. 1B shows one point per measurement site. Are these points campaign means? I suggest you add the error bars to these figures and provide information on where kOVOCs, kAVOCs, kBVOCs, and kNO are obtained from. Are all the data from low NO concentrations or just the OVOC panel?

g. You state many places throughout the manuscript that they measure radical concentrations, but do not specify which radicals they measure.

h. The laser-induced fluorescence system (LIF) based on fluorescence assay by gas expansion technique does not have a reference.

8. I suggest some grammar checks for the manuscript in addition to improving the clarity and precision of the text. At its current state, I do not believe the methodology is reproducible.

References:

Iyer, S., Kumar, A., Savolainen, A., Barua, S., Daub, C., Pichelstorfer, L., Roldin, P., Garmash, O., Seal, P., Kurtén, T., and Rissanen, M.: Molecular rearrangement of bicyclic peroxy radicals is a key route to aerosol from aromatics, *Nat Commun*, 14, 4984, <https://doi.org/10.1038/s41467-023-40675-2>, 2023.

Wang, S., Wu, R., Berndt, T., Ehn, M., and Wang, L.: Formation of Highly Oxidized Radicals and Multifunctional Products from the Atmospheric Oxidation of Alkylbenzenes, *Environ. Sci. Technol.*,

51, 8442–8449, <https://doi.org/10.1021/acs.est.7b02374>, 2017.

Wang, Z., Ehn, M., Rissanen, M. P., Garmash, O., Quéléver, L., Xing, L., Monge-Palacios, M., Rantala, P., Donahue, N. M., Berndt, T., and Sarathy, S. M.: Efficient alkane oxidation under combustion engine and atmospheric conditions, *Commun Chem*, 4, 1–8, <https://doi.org/10.1038/s42004-020-00445-3>, 2021.

Response to Reviewer #1:

General comments:

This work addresses the issue of the underestimation of OH concentrations by theoretical atmospheric models in the low NO and high VOC regime.

Although my own work lies within the atmospheric chemistry field, I was actually unaware of this very interesting problem. Because of this, I cannot give a full account on how much effort has been put over the years into clarifying this divergence between experimental observation and theoretical modeling. However, the authors did a great job in contextualizing and explaining the scientific problem, revealing that the scientific community still has an incomplete understanding (at least until now, in this reviewer's perspective) of the chemistry behind OH regeneration.

The authors tackle this problem by performing a retrospective meta-analysis of a comprehensive warm-season field observation dataset and also by performing a theoretical study, discovering that the autoxidation of carbonyl RC(O)O_2 radicals derived from higher aldehydes could prove to be a novel and noteworthy (again, in this reviewer's eyes) OH regeneration mechanism that could explain the missing source of OH radicals.

As a theoretical and computational chemist I prefer not to make comments about the experimental observations. However, I can provide some comments on the theoretical calculations.

The authors start with hexanal, a C6 aldehyde, which suffers oxidation mainly by OH and then reacts with O_2 to yield the respective R(CO)O_2 radical. The latter radical then undergoes an H-migration reaction, for which its rate constant is calculated with multi-structural transition state theory (MS-TST). The choice of MS-TST is completely adequate for this problem, as also the level of theory used for the respective electronic structure calculations that support this rate constant, based on their careful benchmark calculations. However, the Supplementary Information (SI) does not provide enough information to make a full analysis or reproduction of the calculations. I believe that, at this stage, this manuscript lacks technical information in order for it to be more consistent and for it to better support the conclusions made by the authors. The authors may consider the following suggestions/questions.

Response: Thanks for your constructive comments. We have taken all these suggestions into account and have made corrections in this revised manuscript. Below are our responses to the specific comments and revised our manuscript and Supplementary Information.

Specific comments:

Q1. The total number of calculated conformers (reactants and transition states, TSs) and their relative energies (in a separate Table) should be mentioned somewhere in the "Text S3" part of the SI. Figure S6 only shows one reactant R(CO)O_2 conformer, but there should be much more, given the number of rotatable bonds in this molecule.

Response: Thank you for your comment! We have added the calculated details for searching the conformers in "Text S3" part of the SI. The original Supplementary Fig. 6 is Supplementary Fig. 6A

in the revised manuscript because we also added Supplementary Fig. 6B.

The additions to Supplementary Text 3 are shown below:

For the reactant $\text{CH}_3\text{CH}_2\text{CH}_2\text{CH}_2\text{CH}_2\text{C}(\text{O})\text{O}_2$, we considered five rotation bonds that are C1-O6, C1-C3, C3-C8, C8-C11, and C11-C14, which produce 63 distinguishable conformers. However, for the transition state TS2, there is only one rotation bond C11-C14, producing 3 distinguishable conformers. For the transition state TS4, there are two rotation bonds C8-C11 and C11-C14, producing 8 distinguishable conformers. For the transition state TS5, there are three rotation bonds C3-C8, C8-C11, and C11-C14, producing 21 distinguishable conformers. There are no rotation bonds for TS1 and TS3. The molecules of reactant and transition states for producing conformers are plotted in Supplementary Fig. 6B. The relative energies of the distinguishable conformer of the reactants and transition states are listed in Supplementary Table 8. The conformers shown in Supplementary Fig. 6A are the lowest energy structures for both the reactant and transition states. Extra information on vibrational frequency, absolute energies, and cartesian coordinates are shown in Supplementary Tables 10-12.

The revised Supplementary Fig. 6A and 6B are as follows:

Supplementary Figure 6A. The relative enthalpies at 0 K for the H-migration of $\text{R}(\text{CO})\text{O}_2$ radicals derived from hexanals ($\text{CH}_3\text{CH}_2\text{CH}_2\text{CH}_2\text{CH}_2\text{C}(\text{O})\text{O}_2$). Values (black) are given for all species as calculated by M06-2X/MG3S, and values in parentheses (red) are calculated by WMS//M06-2X/MG3S. It is noted that the conformers for reactants, transition states, and intermediate products are the lowest conformer, respectively.

Supplementary Figure 6B. The molecules of reactant and transition states for producing conformers.

Supplementary Table 8. The relative energies (kcal/mol) of distinguish conformers for reactant and transition states.

R(C ₆ H ₁₁ O ₃)		TS5		TS4		TS2	
structures	Relative energies	structures	Relative energies	structures	Relative energies	structures	Relative energies
1	0.092025	1	0.000182	1	0.000000	1	0.000000
2	0.536567	2	0.000029	2	0.000735	2	0.001698
3	0.507323	3	0.000000	3	0.001527	3	0.001296
4	0.555868	4	0.001566	4	0.000760		
5	2.294800	5	0.000528	5	0.002054		
6	0.000000	6	0.000935	6	0.001400		
7	0.49538	7	0.002271	7	0.003152		
8	0.459814	8	0.001429	8	0.002268		
9	0.608492	9	0.001644				
10	0.607869	10	0.001391				
11	2.12493	11	0.001330				
12	1.27288	12	0.002840				
13	0.931321	13	0.001930				
14	1.39412	14	0.003220				
15	1.38061	15	0.004947				

16	1.30647	16	0.001521				
17	0.808449	17	0.001516				
18	2.5835	18	0.001459				
19	1.35445	19	0.002200				
20	3.2044	20	0.003185				
21	1.29774	21	0.002741				
22	0.855578						
23	1.26603						
24	1.3727						
25	0.644786						
26	0.744275						
27	3.02219						
28	1.84633						
29	1.73491						
30	1.29639						
31	1.8373						
32	1.77948						
33	0.949366						
34	0.732864						
35	1.17509						
36	1.14600						
37	1.19479						
38	2.93461						
39	0.530398						
40	1.02000						
41	0.98994						
42	1.07411						
43	1.06325						
44	2.68009						
45	1.32161						
46	1.52261						
47	1.97909						
48	1.97021						
49	1.90905						
50	1.78167						
51	3.56164						
52	1.93452						
53	3.77096						
54	1.88394						
55	1.22678						
56	1.6589						
57	1.73196						
58	1.10393						

59	1.17089						
60	3.46248						
61	1.91493						
62	1.96923						
63	1.14423						

Note: This table marked in bold font is the lowest energy structure, which is set to the energy zero, and the energy of other independent structures is obtained relative to the lowest energy structure. All conformers of the reactant (R) were optimized using M06-2X/MG3S method. Due to the higher barrier of TS2, TS4, and TS5, they hardly effect the kinetics of the reaction. Thus, all conformers of transition states (TS2, TS4 and TS5) were optimized using the more economical B3LYP/6-31G (d, p) method. Then, all lowest energy conformers of reactant and transition states were reoptimized using M06-2X/MG3S method to obtain the zero-point vibrational energy correction.

Q2. Do the five shown TSs all connect to that one specific reactant conformer as shown in Figure S6?

Response: Yes, we had done the IRC calculation to verify the TSs correctly connect to the specific reactant conformer. Moreover, we have provided IRC calculated results as shown in Supplementary Figs. 14-18. However, in Supplementary Fig. 6A of the revised article, we only consider the lowest reactant conformer for the reactant. In the revised article, we have added some comments in original Line 133: It is noted that the conformers for reactants, transition states, and intermediate products are the lowest conformer, respectively. The original Supplementary Fig. 6 is Supplementary Fig. 6A in the revised article and we also added Supplementary Fig. 6B.

The revised Supplementary Figs. 14-18 are shown below.

Supplementary Figure 14. Intrinsic Reaction Coordinate (IRC) for TS1 (1,7-H migration).

Supplementary Figure 15. Intrinsic Reaction Coordinate (IRC) for TS2 (1,6-H migration).

Supplementary Figure 16. Intrinsic Reaction Coordinate (IRC) for TS3 (1,8-H migration).

Supplementary Figure 17. Intrinsic Reaction Coordinate (IRC) for TS4 (1,5-H migration).

Supplementary Figure 18. Intrinsic Reaction Coordinate (IRC) for TS5 (1,4-H migration).

Q3. Are there only five TSs or these are only the lowest energy ones?

Response: We considered five H-migration sites along the C-C skeletal chain of the reactant $\text{CH}_3\text{CH}_2\text{CH}_2\text{CH}_2\text{CH}_2\text{C}(\text{O})\text{O}_2$, which produces five transition states, as plotted in Supplementary Fig. 6A. All TSs are the lowest energy conformers. The original Supplementary Fig. 6 is Supplementary Fig. 6A in the revised article and we also added Supplementary Fig. 6B.

Q4. Please provide Cartesian coordinates for all calculated reactant and TS conformers.

Response: Thanks for your helpful suggestions. The optimized geometries of all calculated reactants and TS conformers are provided in Supplementary Table 12 of the Supplementary Information.

The newly added Supplementary Table 12 is shown below:

Supplementary Table 12. Cartesian coordinates (Å) for all reactants and transition states are calculated

(a) by CCSD(T)-F12a/cc-pVDZ-F12 method

Species	Coordinates (Å)			
$\text{CH}_3\text{C}(\text{O})\text{OO}$	C	-0.5472837715	0.1343811967	-0.0000018576
	O	-1.4797667421	0.8715784358	-0.0000060715
	C	-0.5321813765	-1.3609565398	-0.0000018102
	H	-1.5643378934	-1.6970387414	-0.0000042745
	H	-0.0050105421	-1.7311039463	-0.8772616845
	H	-0.0050144348	-1.7311042073	0.8772602506
	O	0.7367463172	0.7769653233	0.0000039579
	O	1.7467394433	-0.0807555210	0.0000064899
TS	C	0.1074891169	1.3458731048	-0.0187056285
	H	-1.1163153873	0.9806530913	-0.0403509646
	H	0.3482079524	1.8932090218	-0.9242817433
	H	0.3039176970	1.8938713082	0.8973074132
	C	0.5843008678	-0.0793880365	-0.0043770128

	O	-0.4849500030	-0.9363390581	-0.0023173767
	O	1.7087998615	-0.4755471434	0.0047358132
	O	-1.6850024654	-0.1449671081	-0.0253358105
P	C	0.4685429295	0.1289102180	-0.0000006248
	O	0.2183432333	1.3201666146	0.0000090735
	C	1.7754899264	-0.4841992210	-0.0000017314
	H	2.6369716088	0.1633739607	0.0000058395
	H	1.8752184156	-1.5581719324	-0.0000100672
	H	-1.5226455329	0.7684951619	0.0000063108
	O	-0.5126021178	-0.8114127035	-0.0000118428
	O	-1.7983924629	-0.1747030981	-0.0000049579

(b) by M06-2X/MG3S method

Species	Coordinates (Å)			
CH ₃ C(O)OO	C	-0.54398000	0.13244000	0.00000000
	O	-1.46787700	0.86678400	0.00000400
	C	-0.53391600	-1.35931300	-0.00000200
	H	-1.56504000	-1.69407600	-0.00001200
	H	-0.00640500	-1.73085600	-0.87593500
	H	-0.00642400	-1.73085800	0.87594200
	O	0.73579700	0.76333200	-0.00000800
	O	1.73773600	-0.06548700	0.00000600
TS	C	-0.58264300	-0.07558000	0.00000000
	O	-1.69930000	-0.46687100	0.00000000
	C	-0.10058300	1.34727000	0.00000000
	H	-0.31006800	1.89748200	0.91049200
	H	1.13074300	0.96474600	0.00000000
	H	-0.31006800	1.89748200	-0.91049200
	O	0.48524300	-0.92129300	0.00000000
	O	1.66265100	-0.16056700	0.00000000
P	C	0.47521700	0.13536800	0.00000100
	O	0.23883500	1.32166800	-0.00000200
	C	1.77382700	-0.48517700	-0.00000200
	H	2.63859100	0.15600400	0.00002200
	H	1.86847900	-1.55826600	-0.00003000
	H	-1.56002400	0.76216100	0.00000000
	O	-0.51215000	-0.78938900	0.00000600
	O	-1.78184900	-0.18991000	-0.00000300
C ₆ H ₁₁ O ₃	C	1.93363100	0.50842200	0.00000000
	O	2.00225900	1.68737600	0.00000000
	C	0.71317200	-0.35664200	0.00000000
	H	0.76473700	-1.01783300	-0.86756900
	H	0.76473700	-1.01783300	0.86756900
	O	3.19051100	-0.17040200	0.00000000
	O	3.09777900	-1.46729200	0.00000000

	C	-0.55795400	0.47962900	0.00000000
	H	-0.55920800	1.13584000	-0.87246400
	H	-0.55920800	1.13583900	0.87246400
	C	-1.80898200	-0.38994200	0.00000000
	H	-1.79837000	-1.04683700	0.87547000
	H	-1.79837000	-1.04683600	-0.87547100
	C	-3.09551400	0.42760400	0.00000000
	H	-3.10324600	1.08285900	-0.87457700
	H	-3.10324600	1.08285900	0.87457600
	C	-4.34260300	-0.44800400	0.00000000
	H	-4.36525300	-1.09141700	0.88052500
	H	-5.25221400	0.15092200	0.00000000
	H	-4.36525300	-1.09141700	-0.88052500
	C	-1.61282600	-0.28607900	-0.13668400
	O	-2.48669500	-0.18715200	-0.93921100
	C	-1.23405300	0.76189200	0.87448900
	H	-2.15160500	1.27548200	1.14949000
	H	-0.79915200	0.31077300	1.76029200
	O	-0.81145100	-1.40535700	-0.23296900
	O	0.06310600	-1.57796300	0.81965000
	C	-0.25919300	1.76393800	0.21996400
	H	0.12389000	2.42307900	1.00048900
	H	-0.82183800	2.38427200	-0.47756200
	C	0.90602500	1.12016500	-0.53381800
	H	1.54495900	1.91525600	-0.93604300
	H	0.53211200	0.57736200	-1.40823200
	C	1.76355300	0.19405400	0.28977100
	H	0.96866100	-0.77142900	0.61017700
	H	2.01619900	0.59043400	1.27517800
	C	2.91016600	-0.46473500	-0.42884400
	H	3.64150400	0.28707900	-0.74136500
	H	3.42317400	-1.18648600	0.20370900
	H	2.56038300	-0.97745400	-1.32515100
TS1	C	1.53088100	0.39740800	-0.27284700
	O	2.10944000	0.74922400	-1.25218900
	C	1.54102600	-0.98697300	0.31672600
	H	2.38237200	-1.50664700	-0.13236700
	H	1.68339400	-0.92484200	1.39292000
	O	0.70895100	1.32193300	0.33349200
	O	0.13682600	0.87355000	1.50807900
	C	0.22748200	-1.74481300	0.00414500
	H	0.26261200	-2.69311500	0.53774100
	H	0.21548700	-1.97312800	-1.06473400
	C	-1.03709300	-0.97633700	0.35153100
TS2				

	H	-0.62194700	-0.02582700	1.11099300
	H	-1.72757200	-1.50024800	1.01182800
	C	-1.71582000	-0.25009300	-0.78257300
	H	-2.14456900	-1.00651600	-1.45168800
	H	-0.96879400	0.28369500	-1.37617100
	C	-2.79807900	0.71312400	-0.31255900
	H	-2.36289800	1.49836000	0.30563900
	H	-3.29890100	1.18341200	-1.15662400
	H	-3.55131300	0.19330800	0.28087800
	C	1.49427700	-0.02954900	-0.07346900
	O	2.28638000	-0.53935500	-0.80274500
	C	0.74108800	-0.75676500	1.01021000
	H	1.49253000	-1.26856900	1.60997200
	H	0.19571300	-0.07455800	1.65241800
	O	1.22074300	1.29956200	-0.30829500
	O	0.36215200	1.88422700	0.59847500
	C	-0.18609700	-1.79665700	0.36557000
	C	-1.21529700	-1.21309600	-0.60337100
	H	-1.78463700	-2.04446900	-1.01866900
	H	-0.69800200	-0.75235500	-1.45035400
	C	-2.18250200	-0.19415000	0.02571100
	H	-2.22631000	-0.33118600	1.10954600
	H	-3.19885600	-0.39661600	-0.33131900
	C	-1.89337500	1.24574100	-0.30271800
	H	-1.72651200	1.45416900	-1.35800600
	H	-2.54703700	1.97569700	0.16904500
	H	-0.73655800	1.60441800	0.18790200
	H	0.42918800	-2.52274300	-0.16599900
	H	-0.70228900	-2.33240600	1.16438600
TS3	C	1.49427700	-0.02954900	-0.07346900
	O	2.28638000	-0.53935500	-0.80274500
	C	0.74108800	-0.75676500	1.01021000
	H	1.49253000	-1.26856900	1.60997200
	H	0.19571300	-0.07455800	1.65241800
	O	1.22074300	1.29956200	-0.30829500
	O	0.36215200	1.88422700	0.59847500
	C	-0.18609700	-1.79665700	0.36557000
	C	-1.21529700	-1.21309600	-0.60337100
	H	-1.78463700	-2.04446900	-1.01866900
	H	-0.69800200	-0.75235500	-1.45035400
	C	-2.18250200	-0.19415000	0.02571100
	H	-2.22631000	-0.33118600	1.10954600
	H	-3.19885600	-0.39661600	-0.33131900
	C	-1.89337500	1.24574100	-0.30271800
	H	-1.72651200	1.45416900	-1.35800600
	H	-2.54703700	1.97569700	0.16904500
	H	-0.73655800	1.60441800	0.18790200
	H	0.42918800	-2.52274300	-0.16599900
	H	-0.70228900	-2.33240600	1.16438600
	C	2.25773800	-0.29072000	0.00511700
	O	3.30714100	-0.80640100	-0.20395100
	C	0.95820900	-0.99371300	0.33856500
	H	1.09977200	-2.03919900	0.06720900
	H	0.81548700	-0.93486700	1.42001900
	O	2.16072800	1.07140000	-0.13582900
	O	0.98629000	1.60887900	0.36664500
	C	-0.21545000	-0.35060100	-0.35959700
	H	-0.17877700	-0.42315900	-1.44731700
	H	0.12475200	0.86992400	-0.14348000
	C	-1.58000200	-0.55200200	0.23795000
	H	-1.55248400	-0.28863700	1.29968000
	H	-1.82347700	-1.62213700	0.19941500
	C	-2.67452700	0.24137800	-0.46813300
	H	-2.68959600	-0.03241100	-1.52568000
TS4	C	2.25773800	-0.29072000	0.00511700
	O	3.30714100	-0.80640100	-0.20395100
	C	0.95820900	-0.99371300	0.33856500
	H	1.09977200	-2.03919900	0.06720900
	H	0.81548700	-0.93486700	1.42001900
	O	2.16072800	1.07140000	-0.13582900
	O	0.98629000	1.60887900	0.36664500
	C	-0.21545000	-0.35060100	-0.35959700
	H	-0.17877700	-0.42315900	-1.44731700
	H	0.12475200	0.86992400	-0.14348000
	C	-1.58000200	-0.55200200	0.23795000
	H	-1.55248400	-0.28863700	1.29968000
	H	-1.82347700	-1.62213700	0.19941500
	C	-2.67452700	0.24137800	-0.46813300
	H	-2.68959600	-0.03241100	-1.52568000

	H	-2.42632900	1.30449300	-0.42694700
	C	-4.04717400	0.00422200	0.14841200
	H	-4.05625400	0.29505600	1.19944000
	H	-4.81727000	0.57839200	-0.36400700
	H	-4.32187000	-1.04986700	0.09286600
TS5	C	-2.12419500	0.50831200	0.08024500
	O	-2.52000700	1.62237600	-0.00618300
	C	-0.79122300	0.03994900	0.60400600
	H	-0.69984900	0.15036500	1.68336600
	H	-1.09006800	-1.17323400	0.38489600
	O	-2.86355200	-0.55716400	-0.33003900
	O	-2.12681500	-1.73117400	-0.11715000
	C	0.41410100	0.47416200	-0.18540300
	H	0.46944200	1.56764900	-0.13058700
	H	0.27020300	0.22670900	-1.23989000
	C	1.71199800	-0.13751700	0.33109800
	H	1.64235900	-1.22811100	0.28691800
	H	1.84009200	0.12297100	1.38591400
	C	2.93071200	0.32535500	-0.45860100
	H	2.99374200	1.41545600	-0.41413900
	H	2.79465600	0.06709800	-1.51162100
	C	4.22470400	-0.28916700	0.06054800
	H	4.18945900	-1.37751800	-0.00049100
H	5.08558000	0.05041100	-0.51326700	
H	4.39078500	-0.02066300	1.10452600	
PI	C	1.59693500	0.08777500	-0.08261700
	O	2.00059900	1.05823800	-0.66590100
	C	0.17261800	-0.23311500	0.25935100
	H	-0.19318100	-0.91745900	-0.51139500
	H	0.15008700	-0.78649600	1.19884800
	O	2.42714800	-0.88374000	0.33574800
	O	3.76526300	-0.62701200	-0.02225500
	C	-0.69820900	1.01627500	0.31701400
	H	-0.61379200	1.55340500	-0.62872500
	H	-0.31908500	1.68608600	1.08967800
	C	-2.16116500	0.67028300	0.59054000
	H	-2.71737600	1.60596500	0.74403900
	H	-2.24425200	0.12502000	1.53727600
	C	-2.79253200	-0.12495200	-0.49920200
	H	-2.50481400	0.10221000	-1.51912400
	C	-4.05200000	-0.87741900	-0.26065300
	H	-4.28576600	-1.55105100	-1.08334300
	H	-3.99170900	-1.46505000	0.65831100
H	-4.91018700	-0.20275700	-0.14107700	

	H	3.69211100	0.23715700	-0.47183000
P2	C	-1.76929000	0.08321700	-0.02945800
	O	-1.97636700	1.26620800	-0.06730700
	C	-0.43435200	-0.60298300	-0.02377100
	H	-0.43568100	-1.34760800	0.77384600
	H	-0.34488000	-1.15772700	-0.96041300
	O	-2.77079700	-0.81280800	0.00450100
	O	-4.03658000	-0.19489100	-0.02647600
	C	0.71574100	0.38245600	0.13461400
	H	0.56929700	0.95174500	1.06297500
	H	0.66732300	1.12983400	-0.66226200
	C	2.04208700	-0.29046800	0.13029900
	H	-3.79969600	0.75221000	-0.05448100
	H	2.10573200	-1.32318400	0.45491500
	C	3.29570000	0.50495700	0.04158400
	H	3.20554200	1.23254100	-0.77201400
	H	3.41739300	1.10984700	0.95189100
	C	4.53815900	-0.35641600	-0.16091400
	H	4.46053600	-0.93279200	-1.08265700
	H	5.43920300	0.25231400	-0.21541000
H	4.65691400	-1.05983100	0.66374200	
P3	C	-1.61494200	0.05114900	-0.12615600
	O	-1.71162400	1.13513000	-0.63521200
	C	-0.38033300	-0.79768000	-0.03155300
	H	-0.39182700	-1.32530300	0.92284400
	H	-0.46537300	-1.56340600	-0.80786300
	O	-2.66987700	-0.56683600	0.43352400
	O	-3.85235700	0.18937300	0.31227800
	C	0.89267100	0.01669800	-0.21511100
	H	0.95266700	0.78052300	0.56290000
	H	0.84262600	0.55204000	-1.16485500
	C	2.13777000	-0.86163200	-0.17808300
	H	2.07321300	-1.61784200	-0.96430700
	H	2.17843800	-1.40218400	0.77258400
	C	3.42793400	-0.05545400	-0.34967500
	H	3.39223800	0.49375900	-1.29382800
	C	3.67950100	0.88225800	0.77641500
	H	4.20979100	1.81007900	0.62957100
	H	3.49086400	0.56552700	1.79291400
	H	4.26615600	-0.75875700	-0.44593000
H	-3.53354100	0.99218400	-0.14377600	

P4	C	1.78403200	0.09109200	0.11359900
	O	1.95207200	1.27428100	0.22773900
	C	0.50148500	-0.66753700	0.31110500
	H	0.49714900	-1.53628900	-0.35243100
	H	0.55233100	-1.08398200	1.32907100
	O	2.78880400	-0.74490400	-0.19715300
	O	4.01810400	-0.06990100	-0.32773600
	C	-0.70327300	0.18308900	0.12970000
	H	-0.58563500	1.25569500	0.19690800
	H	3.75770600	0.85891700	-0.17452800
	C	-2.05607900	-0.42821700	0.18064500
	H	-2.26809300	-0.79226700	1.19826800
	H	-2.08255400	-1.32504200	-0.45075300
	C	-3.16882000	0.52726000	-0.24190300
	H	-3.13803100	1.41141000	0.39891000
	H	-2.97060600	0.87449900	-1.25800600
	C	-4.54653800	-0.11813900	-0.16982900
	H	-4.76870500	-0.44866400	0.84574800
	H	-5.32867600	0.57574600	-0.47427700
	H	-4.60156100	-0.99112500	-0.82160900
P5	C	-1.58791900	-0.10151300	0.12894200
	O	-1.44122600	0.84168400	0.87699700
	C	-0.62419500	-1.13425800	-0.15009700
	H	-0.89027600	-1.89421400	-0.86999500
	H	-3.18866900	1.27960300	0.35662700
	O	-2.73976300	-0.28900600	-0.55871100
	O	-3.67563000	0.72218200	-0.28212300
	C	0.71694700	-1.06918400	0.47210800
	H	1.15506700	-2.06671700	0.52087800
	H	0.62387200	-0.68440700	1.48968200
	C	1.66133400	-0.13634900	-0.31175500
	H	1.21325100	0.85906600	-0.35281300
	H	1.74915800	-0.49000200	-1.34237500
	C	3.04359900	-0.05545700	0.32480400
	H	2.94169500	0.29386700	1.35499300
	H	3.47537300	-1.05809900	0.37831100
	C	3.97820400	0.86964100	-0.44524300
	H	3.57384000	1.88153000	-0.48671800
	H	4.96147400	0.92041600	0.02039400
	H	4.11034700	0.52279800	-1.47083900

Q5. Please provide the value for all imaginary frequencies of the TSs, along with the respective energetic data of the relevant stationary points that are needed to calculate the Eckart tunneling correction.

Response: All imaginary frequencies of the TSs and the energetic data of all calculated reactants and transition states in Supplementary Fig. 6A are provided in Supplementary Table 8 and Supplementary Table 10. The Supplementary Table 8 has been shown in the Response to Q1, and Supplementary Table 10 is shown below.

Supplementary Table 10. Vibrational frequency (cm^{-1}) for all structures under different theoretical methods. The bolded numbers denote the imaginary frequency of the transition state.

(a) calculated by CCSD(T)-F12a/cc-pVDZ-F12 method

Species	Vibrational frequency (cm^{-1})						
CH ₃ C(O)OO	118.43	152.05	336.89	504.53	534.45	563.06	761.40
	1002.58	1050.59	1136.02	1194.22	1409.21	1468.32	1475.77
	1881.09	3074.95	3151.92	3186.53			
TS	1815.76i	145.88	270.38	490.90	532.81	613.42	667.88
	840.93	904.68	1041.46	1067.34	1141.98	1173.91	1426.03
	1742.26	1879.24	3126.03	3214.51			
P	204.35	322.43	327.38	434.75	489.17	615.33	679.71
	746.44	872.48	999.09	1027.95	1289.60	1459.82	1510.84
	1731.97	3185.12	3314.08	3490.86			

(b) calculated by M06-2X/MG3S method

Species	Vibrational frequency (cm^{-1})						
CH ₃ C(O)OO	140.21	200.14	348.10	516.75	551.54	584.76	
	784.68	1008.00	1055.79	1182.72	1281.22	1405.86	
	1467.57	1473.45	1950.29	3098.32	3166.94	3208.06	
TS	1827.93i	138.71	281.18	502.63	544.81	624.08	
	698.30	867.13	1042.44	1043.23	1060.65	1130.32	
	1193.27	1419.29	1767.45	1948.21	3144.17	3232.14	
P	199.67	330.02	331.51	414.80	439.24	620.52	
	697.34	758.01	909.27	1042.39	1102.86	1316.64	
	1461.73	1525.59	1780.85	3204.69	3328.28	3615.92	
C ₆ H ₁₁ O ₃	56.44	62.63	86.76	132.06	140.26	152.35	222.45
	247.88	252.34	399.22	436.61	530.49	577.04	611.47
	730.21	762.48	835.10	853.63	920.91	974.65	
	1037.46	1074.74	1077.80	1096.90	1133.64	1148.63	1227.01
	1268.16	1282.33	1291.01	1332.56	1335.97	1344.89	1399.56
	1415.75	1421.11	1451.70	1489.69	1494.49	1505.97	1507.16
	1516.44	1942.61	3032.71	3051.37	3060.51	3061.42	3067.36
3080.17	3086.73	3101.49	3119.89	3126.68	3137.27		
TS1	1751.91i	58.92	123.64	149.83	182.07	217.26	
	261.71	300.23	357.13	395.30	497.83	508.28	
	553.08	590.62	738.07	768.76	808.85	834.20	861.25

	924.68	955.85	1008.09	1072.00	1084.54	1112.80	1119.05
	1141.02	1170.18	1185.40	1239.76	1249.78	1299.94	1320.80
	1362.18	1384.16	1407.63	1411.47	1454.38	1478.98	1490.45
	1492.84	1495.81	1503.97	1896.80	3023.20	3040.81	3057.05
	3076.46	3090.33	3111.29	3118.37	3126.96	3148.60	3182.49
TS2	1731.10i	40.67	91.92	141.85	170.46	241.67	262.97
	295.04	388.48	404.00	473.48	512.30	582.08	621.67
	731.84	764.84	780.31	864.05	895.37	927.01	955.89
	1011.01	1060.14	1090.78	1097.83	1118.70	1133.09	
	1167.65	1186.04	1236.85	1275.19	1293.97	1313.13	1336.26
	1378.19	1402.46	1419.56	1474.86	1484.15	1494.50	1499.27
	1507.02	1511.33	1897.87	3016.19	3057.38	3064.43	3074.91
	3109.67	3116.15	3128.28	3133.38	3147.59	3175.59	
TS3	1822.09i	67.16	119.09	140.33	173.96	204.58	319.59
	375.53	383.09	442.26	515.42	533.00	586.19	645.54
	707.40	741.87	782.59	838.22	846.22	917.91	981.50
	1002.42	1038.81	1076.67	1109.07	1116.87	1131.03	1147.40
	1211.94	1229.75	1254.08	1278.62	1308.85	1355.50	1375.50
	1388.57	1398.54	1437.15	1459.86	1475.03	1489.83	1500.46
	1504.54	1891.31	3021.07	3050.72	3068.54	3071.75	3098.25
	3106.79	3110.42	3118.14	3180.85	3185.76		
TS4	1734.13i	56.83	71.49	87.02	149.33	167.23	245.49
	247.86	308.15	361.25	461.60	503.80	567.08	605.93
	696.90	740.19	805.42	830.18	867.12	915.84	975.01
	1030.56	1077.71	1083.43	1095.74	1136.15	1139.36	1203.41
	1241.78	1252.07	1260.93	1283.15	1319.09	1332.70	1380.95
	1413.56	1419.78	1455.51	1472.84	1497.44	1507.16	1513.08
	1585.65	1902.68	3006.77	3053.89	3060.01	3063.29	3074.06
	3091.27	3105.86	3129.22	3133.15	3141.63		
TS5	1668.78i	37.85	48.74	95.69	107.07	133.72	186.70
	213.98	248.95	330.04	418.25	522.25	594.02	682.56
	692.89	726.24	757.93	850.18	881.78	931.55	967.18
	1034.26	1042.80	1072.92	1087.05	1135.45	1151.04	1178.49
	1195.11	1241.24	1251.91	1308.61	1326.20	1336.16	1386.15
	1411.05	1419.35	1471.91	1491.72	1501.00	1506.89	1515.01
	1772.06	1941.05	3028.76	3040.58	3053.39	3062.20	3067.69
	3082.48	3097.47	3128.59	3133.83	3139.71		
PI	17.20	48.02	51.69	105.92	140.67	166.36	213.61
	258.13	299.90	353.41	414.91	449.55	458.34	486.58
	617.23	755.47	771.94	850.42	889.55	973.48	982.62
	999.31	1038.12	1056.30	1090.74	1129.79	1152.60	1194.08
	1225.46	1242.89	1304.40	1334.92	1356.11	1408.09	1416.33
	1429.95	1467.68	1476.23	1481.93	1491.60	1502.94	1517.25
	1844.89	2983.65	2997.80	3040.04	3064.70	3073.03	3080.04

	3107.88	3126.94	3134.20	3172.29	3631.35		
P2	13.03	35.73	54.43	77.85	109.52	201.37	211.00
	240.53	284.60	370.10	384.41	405.72	428.48	502.19
	606.00	741.94	756.24	784.61	920.80	980.08	998.39
	1027.72	1053.17	1069.52	1086.19	1139.78	1155.44	1182.88
	1220.05	1272.14	1279.13	1306.77	1352.78	1409.97	1415.82
	1443.36	1466.38	1474.47	1484.27	1504.09	1509.70	1519.45
	1847.05	2980.16	2994.29	3039.77	3063.60	3065.00	3075.72
	3116.46	3131.40	3142.13	3166.80	3635.02		
P3	13.17	59.38	90.11	99.06	134.27	160.90	217.02
	264.31	293.57	366.06	413.80	419.26	476.09	511.16
	615.93	737.62	753.30	828.90	849.70	905.39	977.74
	989.46	1039.00	1061.37	1088.34	1111.97	1134.90	1197.12
	1209.40	1236.03	1298.76	1316.44	1330.72	1371.84	1391.59
	1420.38	1467.44	1468.87	1478.18	1493.42	1508.53	1517.09
	1847.99	2991.77	3041.18	3061.13	3068.45	3073.78	3082.31
	3105.44	3123.66	3166.04	3271.54	3631.98		
P4	21.25	40.90	63.65	84.79	134.81	194.17	228.97
	242.02	283.03	357.20	374.99	407.58	429.43	495.59
	611.69	733.50	752.35	855.78	912.20	917.91	980.89
	1027.75	1056.93	1079.41	1095.36	1139.84	1153.47	1189.14
	1224.83	1251.92	1279.07	1322.89	1350.22	1403.79	1417.40
	1444.67	1450.37	1473.86	1497.84	1505.82	1512.82	1518.38
	1851.93	2964.11	2979.27	3020.87	3056.60	3060.99	3073.15
	3091.80	3126.39	3137.58	3210.91	3635.51		
P5	34.32	59.24	73.57	120.55	166.65	219.33	223.00
	251.66	301.50	400.64	402.63	434.39	516.55	593.79
	699.46	739.98	758.15	810.49	910.66	925.48	988.95
	1002.55	1035.84	1075.28	1112.12	1122.85	1163.09	1221.79
	1255.81	1274.08	1318.96	1334.19	1343.03	1391.93	1416.98
	1459.65	1487.37	1492.07	1502.03	1505.90	1514.65	1523.73
	1770.53	3046.13	3054.39	3061.78	3068.18	3077.95	3097.73
	3120.98	3129.63	3137.31	3228.30	3612.25		

Note: Bold fonts with the suffix i are the imaginary frequency of the transition state.

Q6. Do the dashed lines in Figure S6 mean that there were Intrinsic Reaction Coordinate (IRC) calculations performed? If so, please state this in “Text S3”. If not, how do the authors know that those connections in Figure S6 exist?

Response: We had done IRC calculations for all TSs to verify it connect to the reactant and the products, as shown in Supplementary Figs. 14-18, as shown in the Response of Q2. We have added some comments in Supplementary Text 3: We also did intrinsic reaction coordinate (IRC)^{1, 2, 3} calculations to show that the transition state connects with the corresponding reactant and intermediate

product as listed in Supplementary Figures 14-18. However, in Supplementary Figure 6A, the reactant and intermediate product are both the lowest conformers of the reactants and intermediates. The lowest conformers can facilitate conversion to the specific conformer along internal rotation, which is connected with the corresponding transition state.

Q7. Could the authors specifically state the type of H-migration (1,x) associated to each TS, for clarity purposes?

Response: In Supplementary Fig. 6A, TS1 is 1, 7 H-migration, TS2 is 1,6 H-migration, TS3 is 1,8 H-migration, TS4 is 1,5 H-migration, and TS5 is 1,4 H-migration reactions. We have added some comments in Supplementary Text 3: TS1, TS2, TS3, TS4, and TS5 correspond to 1, 7 H-migration, 1,6 H-migration, 1,8 H-migration, 1,5 H-migration, and 1,4 H-migration processes as described in Supplementary Figure 6A, respectively.

Q8. The Eckart tunneling correction is well known to be overestimated at low temperatures (clear and very recent examples are shown here: DOI: 10.1039/d2ea00164k), so it is possible that some of the values shown in Table S5 of the present manuscript are overestimated. Such a possibility should be mentioned in the “Text S3” part of the SI.

Response: We agree with your comments. We have added some comments and presented the ambient temperatures in our seven campaigns in Supplementary Text 3: Here, we used Eckart for tunneling⁴, which is often overestimated at low temperatures⁵; this may lead to the values (Supplementary Table 4) overestimated at low temperatures. However, field measurements were done at room temperature or higher. In this study, our campaigns were conducted in summer or autumn without low temperatures, especially the effect of RO₂ autoxidation on radicals focused on noontime and afternoon. To determine whether the temperature was so low that it would overestimate tunneling probability, herein, we present the diurnal ambient temperature in the seven campaigns in Supplementary Figure 12. The daytime temperature varied from 295 K to 307 K, and the variations of temperature during 10:00-16:00 local time were 303-306 K, 299-303 K, 300-304 K, 298-300 K, 298-301 K, 299-301 K, and 300-306 K at Backgarden, Yufa, Wangdu, Heshan, Taizhou, Shenzhen, and Chengdu, respectively. Therefore, the present results are still reliable.

Supplementary Figure 12. The diurnal ambient temperature in the seven warm-season campaigns

Q9. The previous point leads into this next one: how exactly were the RACM2 simulations performed? If the reader, like me, does not know anything about this simulation procedure, then some more details should be given. For example, at what temperature was the simulation performed? Or was it performed at several temperatures? This is important because of the concerns raised in the previous point, which could undermine some of the conclusions of this paper. I assume, however, that because the manuscript refers to “warm-season” several times, the simulation(s) was performed at 298 K, a temperature at which the overestimation of the Eckart tunneling correction is usually not troublesome. If so, please confirm this.

Response: Thank for your helpful suggestions. We have added the description of the model configuration, especially the detailed temperature-related descriptions, as your valuable suggestions in Supplementary Text 2, as shown below: As for the comparison between observed and simulated radical concentrations, a zero-dimensional chemical box model based on RACM2 was utilized in this study. The model was constrained to CO, NO, NO₂, CH₄, O₃, HONO, and VOC concentrations and photolysis frequencies, water vapor, ambient temperature and pressure as well. For RACM2, the measured detailed VOC species were lumped into several categories, which is introduced in previous studies ^{6, 7}. The H₂ and CH₄ mixing ratio was assumed to be 550 ppb and 1900 ppb, respectively. Other parameters were from the observations with 5-min time resolution, which represents the actual atmospheric environmental conditions every day during campaigns. Given the significance of ambient temperature on the results of quantum chemical calculations, herein, it's worth mentioning that the temperature input in the model was the real ambient temperature observed in campaigns rather than a constant value. Additionally, a spin-up time with 2 days at the beginning of simulations was used to reach steady-state conditions for long-lived species. A fixed dilution equivalent of 24-h lifetime was added in the model to represent the dry deposition. Based on the above configuration in the model, simulated radical concentrations would be obtained and then they will be compared to the observed concentrations to investigate whether the chemical mechanisms are correct, which is the most direct indication to identify whether the chemical mechanisms in the troposphere are complete or not.

Besides, the detailed temperature-related descriptions have been added in Supplementary Text 3, as shown in the Response to Q8.

Q10. Please cite the original RACM2 paper in the main manuscript, <http://dx.doi.org/10.1016/j.atmosenv.2012.11.038>

Response: Thanks, we have cited it in the manuscript and Supplementary Information.

Q11. The comments made above are mainly related to the first H-migration reaction of Figure 2A in the main manuscript. The authors then support the remaining reactions of Figure 2A with references. Specifically, in Table S6, they claim that “The rate constants of Nos. 3-5 were from Wang et al.”, which for what I could understand is reference 25 of the SI: <https://pubs.acs.org/doi/10.1021/acs.est.7b02374> However, I could not find the rate constant values

for reactions 3-5 of Table S6 in this reference 25 paper. Could the authors provide a clear explanation of how they came up with these numbers? This is very important in order to make a good case for their conclusions.

Response: Thanks for your helpful comments. We really did not make it clear where the reaction rates came from in original Table S6 (Now is Supplementary Table 5). The rate constants of Nos. 3-5 are from Wang et al. (2019) (DOI:10.1063/1674-0068/cjcp1811265), which was shown in reference 28 in the original manuscript titled “Role of Hydrogen Migrations in Carbonyl Peroxy Radicals in the Atmosphere”⁸. The rate constants of Nos. 3-4 come from the picture of “Scheme 1 -- New pathways of CH₃CH₂CH₂C(O)O₂ radical after H-migration”. As for No. 5, Wang et al. (2019) did not give the rate constant directly. However, as mentioned by Wang et al. (2019)⁸, the RO radicals could react with O₂ at effective rates of (0.5~5) ×10⁴ s⁻¹ with rate coefficients of 10⁻¹⁵~10⁻¹⁴ cm³ molecules⁻¹ s⁻¹ according to the studies from Atkinson et al. (1997) ([https://doi.org/10.1002/\(SICI\)1097-4601\(1997\)29:2<99::AID-KIN3>3.0.CO;2-F](https://doi.org/10.1002/(SICI)1097-4601(1997)29:2<99::AID-KIN3>3.0.CO;2-F))⁹. The radical in No. 5 is α-hydroxy radicals (R₁R₂COH), and Atkinson (1994) ([https://doi.org/10.1016/0960-1686\(90\)90438-S](https://doi.org/10.1016/0960-1686(90)90438-S))¹⁰ once reported the reactions of O₂ and the simplest α-hydroxy radicals (CH₂OH) with the *k*(CH₂OH+O₂) is 9.0×10⁻¹² cm³ molecules⁻¹ s⁻¹ at 298 K, which is larger than the rate constant of RO+O₂. The rate coefficients of CH₂OH+O₂ reactions reported in previous studies were summarized on the IUPAC website (https://iupac-aeris.ipsl.fr/show_datasheets.php?category=Gas-phase+organics%3A+R_oxygen) and were all larger those that of RO+O₂ reactions. To obtain a conservative result, we chose the rate constant in No. 5 as 10⁴ s⁻¹. Herein, we added details about the provenance of these reaction rates in the revised manuscript and Supplementary Table 5 in the new version.

The revision of the notes in the new Supplementary Table 5 is:

Note that:

- (1) ACO₃ and HKET denote acetyl peroxy radicals and hydroxy ketone, respectively. *j*MACR represents the photolysis rate constant of methacrolein (MACR).
- (2) The rate constant of No. 1 is from our quantum chemical calculations.
- (3) The rate constant of No. 2 is from Table 1 of Tan et al.¹¹.
- (4) The reactions of Nos. 3-5 are from Wang et al.⁸ and the rate constants of both No. 3 and No. 4 are from Wang et al.⁸.
- (5) As for No. 5, Wang et al.⁸ did not give the rate constant directly. However, as mentioned by Wang et al.⁸, the RO radicals could react with O₂ at effective rates of (0.5~5) ×10⁴ s⁻¹ with rate coefficients of 10⁻¹⁵~10⁻¹⁴ cm³ molecules⁻¹ s⁻¹ according to the studies from Atkinson⁹. The radical in No. 5 is α-hydroxy radicals (R₁R₂COH), and Atkinson¹⁰ once reported the reactions of O₂ and the simplest α-hydroxy radicals (CH₂OH) with the *k*(CH₂OH+O₂) is 9.0×10⁻¹² cm³ molecules⁻¹ s⁻¹ at 298 K, which is larger than the rate constant of RO+O₂. The rate coefficients of CH₂OH+O₂ reactions reported in previous studies were summarized in IUPAC website (https://iupac-aeris.ipsl.fr/show_datasheets.php?category=Gas-phase+organics%3A+R_oxygen) and were all larger those that of RO+O₂ reactions. To obtain a conservative result, we chose the rate constant in No. 5 as 10⁴ s⁻¹.
- (6) The photolysis rate of No. 6 is a conservative result from Liu et al.¹².

Response to Reviewer #2:

General comments:

Hydroxyl Radicals (OH) is the most important oxidant in the atmosphere, it is critical to understand the atmospheric chemistry as well as the climate, since it determines the fate of almost all air pollutants and reactive greenhouse gases. This study aims to solve a very critical and fundamental problem in the field of atmospheric free radical chemistry, that is, the systematic underestimation of OH radical concentration by existing models under low NO and high VOC regimes. They used the most comprehensive field measurement datasets of atmospheric radical and precursors in the world, as far as I know, to identify the missing source of tropospheric OH radicals, they revealed the higher aldehydes, a typical class of oxygen volatile organic compounds that less concerned in the atmospheric chemistry, play an important role in the regeneration of OH. The new regeneration of OH mechanism by reactive aldehydes is clarified by a series of quantum chemical calculations and well assessed by the field datasets. In the end, they also argued that this new chemistry mechanism proposed in the study would be more important in the low NO_x conditions in the future.

The methods of data acquisition appear sound to me, conclusions are fully supported by data and well-argued based on relevant literature. I'm convinced and impressed by discussion data from different field campaigns across the world to draw the conclusions, which nicely emphasizes the locally wide spread importance of the findings. Overall, this study is well designed and nicely presented. This novel work is potentially important for understanding the atmospheric radical chemistry, and the article will attract widespread attention and have a far-reaching impact in the community of atmospheric and climates sciences, and beyond. Thus, I would like to recommend this interesting article for publication after addressing the minor comments shown below.

Response: Thanks for your constructive comments. We have taken all these suggestions into account and have made corrections in this revised manuscript. Below are our responses to the specific comments and revised our manuscript and Supplementary Information.

Specific comments:

Q1. The first part of the Main Text seems a little confused. The background is introduced and followed by the meta-analysis of the field OH budget. I suggest the author adding a section header to distinguish the background (or Introduction) from the analysis of OH missing source. And in the end of the introduction section, a brief introduction about the method and purpose of this study should be outlined.

Response: Thanks. We adjusted the structure of the manuscript based on the formatting requirements of this journal and this suggestion accordingly. We added one section "Introduction" without subheadings according to the requirements, and move the meta-analysis part to the first part of the Result section "Characterization of the OH missing source."

In the last of the "Introduction", we added a paragraph to introduce the main content of this study: Based on the combination of quantum chemical calculations and observation-constrained model simulations, we report the detailed reaction processes of the autoxidation mechanism of high aldehydes

and further confirm its importance by examination in numerous field studies worldwide.

Q2. This paper proposed the OH regeneration mechanism by the quantum chemical calculations method, we understand that the fine quantification of these key kinetic parameters of higher aldehydes chemistry is a very long-term and large project, it would be too harsh to ask for achieving it in this study. But a clearer outlook should be provided to direct the laboratory studies in the future. I would like to see the author add some discussions in the last paragraph.

Response: Thanks for your suggestions, we added the further potential directions of RO₂ autoxidation in the last paragraph: For example, the kinetic parameters of RO₂ autoxidation from high aldehydes, involving rate constants and product yields, and the photolysis process of hydroperoxyl-carbonyls are need to be further quantified through both quantum chemical calculations and kinetic experiments.

Q3. Line203-210, the explanation of high HPC yield seems to be plausible, but related references should be added to support this argument, especially the cooking emission source of high aldehydes. I suggest the author add the temperature information into the Table S7 to clarify the relationship of temperature and the HPC yield.

Response: Thanks for your suggestions. We have added several papers related to high aldehydes from cooking emission in original Lines 165-174: The presence of still unexplained OH sources after including HAM in these campaigns may be tightly related to the following factors: 1) aldehydes from primary emissions, especially aldehydes with high carbon numbers from biomass burning^{13, 14} or cooking activities^{15, 16}, are not considered in the model, which are likely to be oxidized into RO₂ radicals with much higher H-migration rate than those currently used in this study.

The references related to biomass emission are:

- S. Inomata et al., Laboratory measurements of emission factors of nonmethane volatile organic compounds from burning of Chinese crop residues. *Journal of Geophysical Research-Atmospheres* 120, 5237-5252 (2015). DOI: 10.1002/2014JD022761.
- Y. Zhang et al., Emission inventory of carbonaceous pollutants from biomass burning in the Pearl River Delta Region, China. *Atmospheric Environment* 76, 189-199 (2013). DOI: 10.1016/j.atmosenv.2012.05.055.

The references related to cooking emissions are:

- A. Atamaleki et al., Emission of aldehydes from different cooking processes: a review study. *Air Quality Atmosphere and Health* 15, 1183-1204 (2022). DOI: 10.1007/s11869-021-01120-9.
- R. Dominguez, M. Gomez, S. Fonseca, J. M. Lorenzo, Effect of different cooking methods on lipid oxidation and formation of volatile compounds in foal meat. *Meat Science* 97, 223-230 (2014). DOI: 10.1016/j.meatsci.2014.01.023.

Additionally, according to the suggestions of the reviewer #3, we optimized the derivation of OH yields from the VOC oxidation through HPC photolysis, as shown in the revised Supplementary Table 6, and the ambient temperatures of our seven campaigns were added into this table as well.

Supplementary Table 6. The ϕ values achieving optimized agreement between the observed and modeled OH.

Sites (Campaigns)	ϕ	Ambient temperature (K)
Backgarden (PRIDE-PRD2006)	2.2	306.5 ^a
Yufa (CAREBeijing2006)	2.2	303.0 ^a
Wangdu (NCP2014)	1.5	304.2 ^a
Heshan (PRIDE-PRD2014)	1.1	300.1 ^a
Taizhou (EXPLORE-YRD2018)	1.4	301.4 ^a
Shenzhen (STORM2018)	0.4	300.5 ^a
Chengdu (CHOOSE2019)	1.6	306.0 ^a
Michigan (PROPHET)	2.6	298.1 ^b
Suriname (GABRIEL_daytime)	2.93	296.1 ^c
Suriname (GABRIEL_afternoon)	2.6	296.1 ^d
Borneo rainforest (OP3-I)	1.75	300.1 ^e
New York City (PMTACS-NY2001)	0.68	301.7 ^f
Tokyo (IMPACT-L)	0.6	304.3 ^f
California (BEARPEX09)	0.2	299.1 ^g
London (ClearfLo)	0.45	300.6 ^h

Note that:

^a Max diurnal temperature for noontime

^b Mean temperature for the time window 10:00-11:00 am

^c Mean temperature for daytime (08:00-17:00)

^d Mean temperature for afternoon (14:00-17:00)

^e Mean temperature for noon time (11:00-12:00)

^f Mean temperature for noon time (11:00-13:00)

^g Mean temperature for daytime (09:00-15:00)

^h Max noontime temperature during easterly flow

Q4. Line 165-174, why the HAM can largely explain the missing source but still cannot well address this issue in Backgarden, Yufa, Heshan, and Chengdu? Are there air mass conditions differences or something else, more discussions should be added to address this point.

Response: Thanks for your suggestions. We added some discussions in original Line 174 as following: The presence of still unexplained OH sources after including HAM in these campaigns may be tightly related to the following factors: 1) aldehydes from primary emissions, especially aldehydes with high carbon numbers from biomass burning^{13, 14} or cooking activities^{15, 16}, are not considered in the model, which are likely to be oxidized into RO₂ radicals with much higher H-migration rate than those currently used in this study; 2) the total concentrations of HPC groups were underestimated as they can also be produced from many other VOCs than higher aldehydes (*e.g.*, long-chain alkanes¹⁷, ethers¹⁸, alcohols¹⁹, alkenes²⁰); 3) the underestimation of HPC photolysis rate for HPC species with different functional groups.

Response to Reviewer #3:

General comments:

The authors propose that higher aldehydes represent a significant OH source in the atmosphere, under relatively low nitrogen oxide concentrations. They show that the ratio between the observed and modeled OH concentration exceeds unity at low NO concentrations in seven measurement locations within China that serve as the key observational data set for this study. The authors further suggest that the unexplained OH production rate correlates with the ratio between NO and OVOC concentration. Based on a previous quantum chemical calculation study, they formulate a hypothesis that atmospheric aldehydes, comprising five or more carbon atoms, undergo autoxidation yielding a hydroperoxyl-carbonyl (HPC) and an HO₂ radical. They note, based on one another theoretical study, that HPC can undergo rapid photolysis yielding an OH radical. The authors perform estimations on the H-migration speeds for higher aldehydes (definition is unclear) using 1,7 H-migration speeds from quantum chemical calculations performed for R(CO)O₂ radicals derived from hexanal between 200 and 340 K ($\Delta T = 20$ K). They further parameterized the higher aldehydes autoxidation mechanism (termed HAM) into the RACM2 model and evaluated the contribution of HAM to the observed radical chemistry. The base case set for RACM2 included the well-established OH production rates from HONO and ozone photolysis, alkene ozonolysis, and the HO₂+NO and HO₂+O₃ reactions. In addition, it includes H-migration from isoprene-derived RO₂s (LIM).

The classical OH production channels could explain 15–68% of the OH production (POH) across the seven sites. The importance of the HAM in explaining the remaining OH production seemed to be highly pronounced for the measurement sites, where the classical OH production channels already explained the majority of the OH. In contrast, the modelled importance of HAM was lowest in the sites where OH production could not be reproduced with the classical production channels. HAM was reported as more critical than LIM in OH production across the sites.

Next, the authors suggest (with support from a number of references) that certain VOC species can produce HPC-like reactive aldehydes via autoxidation. As the molar yields for such aldehydes remain unresolved, the authors derive the required HPC yields to bridge the gap between the modeled and measured OH concentrations through sensitivity studies. The molar yields range from 0.1 to 35 for the different sites, and the highest yields correspond to the measurement locations, where HAM showed little relevance in OH production in the previous RACM2 simulations, and the unexplained OH production fraction was the highest. The authors suggest that the need for the very high HPC yield for these measurement sites could arise from temperature effects on HPC formation as the two sites experienced high temperatures albeit based on the shown data in the SI, the temperature differences were not significant.

The new mechanism including the specified VOC autoxidation to HPC with the determined HPC yields is then included in RACM2 (RAM simulations). Because of the fixed yields, the model evaluation could not be performed via OH concentration comparisons, as the “fitted” HPC yields would guarantee a perfect agreement. Instead, the authors evaluate the modeled and observed OH

recycling probabilities and state that a discrepancy that exists in the RACSM2 base case is now eliminated. The authors finally suggest that the HAM mechanism is important over a wide range of NO concentrations and could be highly important in the future due to emission regulations yet the LIM mechanism dominates in the cleanest environments.

Overall, I do find the manuscript interesting and also interesting to a broader audience as understanding the OH budget is crucial from both climate and air quality perspectives from multiple angles. The proposed idea is very good and exciting, and the autoxidation of aldehydes has also been discussed previously in experimental contexts. However, I am unfortunately not convinced about the presented proof of the hypothesis, the quality of the analysis, and the descriptions of the utilized methodologies. Therefore, I cannot recommend the publication of this paper in its current state in Nature Communications.

I have listed a few suggestions below on how to improve this manuscript:

Response: Thanks for your summary and the critical and constructive comments. We have taken all these suggestions into account and well addressed them in this revised manuscript.

Specific comments:

Q1. Could you perhaps evaluate how reasonable the utilized HPC molar yields are? The average temperatures do not look that different in Table S3. Yufa has a lower temperature than Wangdu, but the HPC yield for Yufa is almost two orders of magnitude higher. Are the HPC yields used as constants per measurement sites as shown in Table S7?

Response: Previous studies have reported that the HPC species generated from VOC oxidation could vary significantly from each other²¹, and the photolysis rate of different HPC species could also exhibit great differences with up to around 4 folds, as shown in Table 2 reported by Liu et al.¹². These all indicates that the HPC molar yields may highly varied in different air masses.

According to your valuable suggestions, we redefined “HPC molar yields” in the original manuscript and renamed it “OH yields from the VOC oxidation through HPC photolysis” which is denoted by a simple index as ϕ , to characterize the generalized yield of OH regenerated by the oxidation of VOC through the HPC chemistry based on an inversed model. We further clarified its definition in details in the supporting information. In brief, the ϕ including the yields of different HPC species from the oxidation of different VOC species, the yields of OH radicals in photolysis of different HPC species, and the ratio of the HPC photolysis rate to ten times of the MACR photolysis frequency. Ten times MACR photolysis frequency order of about 10^{-4} s^{-1} , which is the base case setup of HPC photolysis rate used in the HAM.

Furthermore, we have improved the inversed model approach for the calculation of ϕ , which avoids the large bias from the using of undue high HPC photolysis rate. In the revised manuscript, we updated the ϕ values in Supplementary Table 6 based on the newly revised approach, which makes more sense compared to those in the original version. More detailed information on the calculation of ϕ has been added into Supplementary Text 5:

To further explore the effects of generalized RO₂ autoxidation on radical chemistry, we made a rule

of thumb assumption named RAM scheme, in which RO₂ from some certain VOCs (alkanes, alkenes, aromatics, *etc.*) could undergo autoxidation process similar to that of RO₂ from aldehydes in HAM scheme. In the RAM scheme, specific VOCs classes were set to generate HPC according to Table 1 of Bianchi et al. ²¹. Here, the specific VOCs include HC5 (alkanes, esters, and alkynes with OH rate constant (298 K, 1 atm) between 3.4×10⁻¹² and 6.8×10⁻¹² cm³ s⁻¹), HC8 (alkanes, esters, and alkynes with OH rate constant (298 K, 1 atm) greater 6.8×10⁻¹² cm³ s⁻¹), OLT (terminal alkenes), OLI (internal alkenes), DIEN (butadiene and other anthropogenic dienes), BEN (benzene), TOL (toluene), XYM (m-xylene), XYP (p-xylene), XYO (o-xylene), ISO (isoprene) in RACM2.

In this scheme, we defined a simple index, named as ϕ , to characterize the generalized yield of OH regenerated by the oxidation of VOC through the HPC chemistry, which is derived from an inversed modeling method, to achieve a good agreement between the modeled and observed OH radicals. The definition of ϕ is shown as follows (Supplementary Equation (5)).

$$\phi = \sum_{j=1}^n (\sum_{i=1}^n A_{ij} \times \text{HPC}_j) \times B_j \times R_j \quad \text{Supplementary Equation (5)}$$

A_{ij} denotes the yields of different HPC_j species from the oxidation of different VOC_i species.

B_j denotes the yields of OH radicals in photolysis of different HPC_j species.

R_j denotes the ratio of the HPC_j photolysis frequency to ten times MACR photolysis frequency. Ten times MACR photolysis frequency order of about 10⁻⁴ s⁻¹, which is the base case setup of HPC photolysis rate used in the HAM.

Where ϕ depends on the values of A_{ij} , B_j and R_j . The HPC species and the HPC yield are heavily dependent on VOC species (A_{ij} , referred to Table 1 of Bianchi et al. ²¹). Subsequently, in the HPC photolysis reaction, different HPC species would have different OH yields (B_j) as well as the photolysis frequencies (R_j). For example, the photolysis frequencies may differ by more than a factor of 4 between different HPC species (referred to Table 2 of Liu et al. ¹²). The three coefficients would be highly varied in different campaigns, which would probably further amplify the differences in the derived ϕ in different campaigns.

Herein, based on the reversed model, we calculated the ϕ value, which is the equivalent mean value for each campaign, in our fourteen campaigns in Supplementary Table 6.

Besides, the ambient temperature might not be the limited factor to influence the OH yields from the VOC oxidation through HPC photolysis (Supplementary Table 6) in RAM, since the daily maximum temperature were over 296 K in our seven campaigns and showed overall small temperature differences. Compared to the ambient temperature, VOC components may be more decisive for OH yields. Despite of the similar ambient temperature at Yufa and Wangdu sites, their reactive VOC species exhibited great different ^{7, 22, 23}. Thus, we updated the explanation for different contributions of RAM to different campaigns in the original Lines 199-210: We defined a simple index, named as ϕ , to characterize the generalized molar yield of OH regenerated by the oxidation of VOCs through the HPC chemistry. The index ϕ is derived from an inversed modeling method, to achieve a good agreement between the modeled and observed OH radicals (Supplementary Text 5). A reasonable ϕ of 0.4-2.2 is derived for our seven warm-season campaigns (Supplementary Table 6). Additionally, we

show the incorporation of RAM into the model would further improve the agreement between the observed and simulated HO₂ concentrations to some extent (Supplementary Fig. 8). The variation of the index ϕ in different campaigns may be caused by: 1) HPC species and HPC yield produced by different VOC oxidation processes may be highly varied²¹. 2) The OH yield from the HPC photolysis may be varied for different HPC species. 3) HPC photolysis frequencies which may differ by over four times¹².

Supplementary Table 6. The ϕ values achieving optimized agreement between the observed and modeled OH.

Sites (Campaigns)	ϕ	Ambient temperature (K)
Backgarden (PRIDE-PRD2006)	2.2	306.5 ^a
Yufa (CAREBeijing2006)	2.2	303.0 ^a
Wangdu (NCP2014)	1.5	304.2 ^a
Heshan (PRIDE-PRD2014)	1.1	300.1 ^a
Taizhou (EXPLORE-YRD2018)	1.4	301.4 ^a
Shenzhen (STORM2018)	0.4	300.5 ^a
Chengdu (CHOOSE2019)	1.6	306.0 ^a
Michigan (PROPHET)	2.6	298.1 ^b
Suriname (GABRIEL_daytime)	2.93	296.1 ^c
Suriname (GABRIEL_afternoon)	2.6	296.1 ^d
Borneo rainforest (OP3-I)	1.75	300.1 ^e
New York City (PMTACS-NY2001)	0.68	301.7 ^f
Tokyo (IMPACT-L)	0.6	304.3 ^f
California (BEARPEX09)	0.2	299.1 ^g
London (ClearfLo)	0.45	300.6 ^h

Note that:

^a Max diurnal temperature for noontime

^b Mean temperature for the time window 10:00-11:00 am

^c Mean temperature for daytime (08:00-17:00)

^d Mean temperature for afternoon (14:00-17:00)

^e Mean temperature for noon time (11:00-12:00)

^f Mean temperature for noon time (11:00-13:00)

^g Mean temperature for daytime (09:00-15:00)

^h Max noontime temperature during easterly flow

Q2. Could explain the methods behind the observed OH recycling probability? If this is used as some sort of validation metric for your proposed hypothesis, I believe it is crucial to explain it thoroughly.

Response: We apologize for the misunderstanding. In fact, the observed OH recycling probability is not a validation metric for our proposed hypothesis but a metric to evaluate the environmental impact of new reaction pathway.

To further clarify the OH recycling probability, we added more detailed description in “Method: Calculation of OH recycling probability” in the revised manuscript:

The stability of OH chemical system mainly could be affected by emissions of NO_x, CO, and CH₄.

The OH recycling would be inefficient under low NO_x levels. When CO and CH₄ are very high and growing rapidly, such conditions can become catastrophic as both O₃ and HO_x are removed. Conversely, when NO_x concentrations are high, the OH recycling is efficient and the system could become autocatalytic, causing a runaway of oxidants. Such high-NO_x conditions would lead to the efficient OH recycling on the one hand, and on the other hand it would lead to an increase in OH removal due to the reaction of OH and NO₂. Herein, to evaluate the effect of the newly proposed HAM on radical chemistry, we utilized the OH recycling probability, calculated from primary OH formation and OH recycling, to reflect the stability of tropospheric hydroxyl chemistry.

OH recycling probability (γ) was once defined by Lelieveld et al., in which both OH primary production rate and secondary production rate are taken into account²⁴. The oxidation power (G) denote the gross OH formation, which is the sum of primary (P) and total secondary (S) OH formation, as shown in Equation (1).

$$G = P + S = P + \gamma * P + \gamma^2 * P + \dots = \frac{P}{1 - \gamma} \quad \text{Equation (1)}$$

Thus, the OH recycling probability is calculated by Eq. S7.

$$\gamma = 1 - \frac{P}{G} \quad \text{Equation (2)}$$

When γ approaches 1, it is indicated that OH radicals are insensitive against primary pathways and are almost entirely from secondary generation pathways. Under such condition, the OH recycling is very efficient and the OH chemistry system could become autocatalytic. The γ being approximately equal to 0.5 denotes the OH formation is quite efficient but not autocatalytic, demonstrating the OH chemical system may be stable. Under low γ conditions, OH radicals are mainly from primary pathways, and thus, greatly sensitive to O₃ photolysis, *etc.* Overall, for the evaluation of newly proposed mechanism, OH recycling probability is a comprehensive indicator from the perspective of primary and secondary generation pathways.

Q3. How does the measured HO₂ concentration compare with your RAM simulation outputs? Are there other ways you could evaluate your model results? How can you be sure that what you propose is actually happening?

Response:

We evaluated the impacts of RAM scheme in HO₂ concentrations accordingly. We found the RAM improve the agreement between the observed and simulated HO₂ concentrations to some extent. Therefore, here we added some discussion into the revised manuscript in original around Lines 200-210: Additionally, we show the incorporation of RAM into the model would further improve the agreement between the observed and simulated HO₂ concentrations to some extent (Supplementary Fig. 8).

Supplementary Figure 8. NO dependence on HO₂ radicals in our seven warm-season campaigns. The red box-whisker plots give the median, the 75th and 25th percentiles, and the 90th and 10th percentiles of the radical observations. The blue circles show the median values of the simulations by the base model, and the green circles show the simulations by the model with RAM. Only daytime values and NO concentration above the detection limit of the instrument were chosen.

(2) As you proposed, HO₂ could be another way to evaluate our model results and showed that our newly proposed mechanism is actually happening. In the future, more direct evidence on HPC generation by direct measurement is needed to prove the happening of RO₂ autoxidation. At the current stage, your recommended paper in Q5²⁵ and other related studies^{21, 26} have provided evidence of RO₂ autoxidation process and the formation of HPC species based on the direct measurements in the laboratory. Besides, Liu et al. have been reported the fast photolysis of HPC yields¹². In summary, we believe that this proposed mechanism could happen in the ambient air. Additionally, as reviewer #2 mentioned, the fine quantification of these key kinetic parameters of higher aldehydes chemistry is a very long-term and large project, it would be too harsh to ask for achieving it in this study. Based on this, we provided a clearer outlook to direct the laboratory studies in the future: **For example, the kinetic parameters of RO₂ autoxidation from high aldehydes, involving rate constants and product yields, and the photolysis process of hydroperoxyl-carbonyls are needed to be further quantified through both quantum chemical calculations and kinetic experiments.**

Q4. Regarding the aromatic autoxidation mechanism discussed in L174–L177, you might be interested in the very recent publication by Iyer et al. (2023) and the discussion related to the findings reported by Wang et al. (2017).

References:

[1] Iyer, S., Kumar, A., Savolainen, A., Barua, S., Daub, C., Pichelstorfer, L., Roldin, P., Garmash, O., Seal, P., Kurtén, T., and Rissanen, M.: Molecular rearrangement of bicyclic peroxy radicals is a

key route to aerosol from aromatics, *Nat Commun*, 14, 4984, <https://doi.org/10.1038/s41467-023-40675-2>, 2023.

[2] Wang, S., Wu, R., Berndt, T., Ehn, M., and Wang, L.: Formation of Highly Oxidized Radicals and Multifunctional Products from the Atmospheric Oxidation of Alkylbenzenes, *Environ. Sci. Technol.*, 51, 8442–8449, <https://doi.org/10.1021/acs.est.7b02374>, 2017.

Response: Thanks for the suggestion. We have carefully studied the above two papers and cited these two papers and updated the description of aromatic autoxidation mechanism, as shown in original Lines 174-177: Additionally, aromatic autoxidation mechanisms have been proposed recently, demonstrating the significant role of OH-initiated oxidation of aromatics in forming highly oxidized products and thus secondary organic aerosols^{27,28}, while the mechanism was found negligible for OH generation in the seven field campaigns (Supplementary Text 4 and Supplementary Fig. 7).

Q5. The discussion about aldehyde autoxidation is also presented in Wang et al. (2021) could perhaps interest the authors of this manuscript.

Reference: Wang, Z., Ehn, M., Rissanen, M. P., Garmash, O., Quéléver, L., Xing, L., Monge-Palacios, M., Rantala, P., Donahue, N. M., Berndt, T., and Sarathy, S. M.: Efficient alkane oxidation under combustion engine and atmospheric conditions, *Commun Chem*, 4, 1–8, <https://doi.org/10.1038/s42004-020-00445-3>, 2021.

Response: Thanks for your suggestions. Your recommended paper would further enrich our understanding of aldehyde autoxidation and further support the significance of aldehyde autoxidation process under atmospheric conditions in our study. We have cited this paper in original Line 127 and Lines 190-192 to support the autoxidation of RO₂ radicals derived from alkanes.

Line 127: Wang et al. further affirmed the high autoxidation potential of aldehydes under atmospheric conditions²⁵.

Lines 190-192: Recent studies have suggested that RO₂ derived from VOCs besides aldehydes, such as alkanes, aromatics, and other OVOCs (ketones, ethers, *etc.*), could also undergo autoxidation with possible subsequent generation of radicals^{17, 18, 25, 28, 29, 30, 31}.

Q6. I would also be curious to hear your motivation for the discussion in L262–L264 presented in your manuscript on the effects of NO on RAM. In the Wang et al. (2021) study, they measure increases in the molar yields of alkane-derived oxygenated products (incl. RO₂) as a function of NO. They suggest that multi-step isomerization of RO and/or RO₂ radicals are likely for all types of alkanes even at high NO concentrations (they went up to roughly 10 ppb). How would accounting for such change your results and the OH_{obs}/OH_{mod} under high NO concentrations?

Response: Thanks for your helpful suggestions and we have studied the research of Wang et al. (2021) which would further enrich our study. This study reported that alkane autoxidation process would also exist even at high NO concentrations. Nevertheless, in theory, although RO₂ autoxidation could occur under high NO conditions, while this process becomes less significant for radical generation under high NO conditions because RO₂ removal is dominated by the reactions with NO under these ambient conditions. Additionally, our study focuses more on radical chemistry under low-

NO conditions, where OH underestimation is more likely occur.

Herein, we evaluated the impact of different NO concentrations on radical generation from RO₂ autoxidation process.

We added this part into Supplementary Text 7 named Influences of HAM in ratios of OH_{obs} to OH_{mod} under different NO concentrations in China.

The main content is as follows: To further explore the influence of HAM in ratios of the observed to modeled OH concentrations under different NO concentrations, we present the NO dependence of OH_{obs}/OH_{mod} at two sensitivity tests involving without and with HAM in the model, as shown in Supplementary Figure 13A. The incorporation of HAM into the model caused the decrease of OH_{obs}/OH_{mod}, indicating that a better agreement between the observed and modeled OH concentrations would be achieved when HAM was considered in the model. Additionally, the decline rates of OH_{obs}/OH_{mod} with HAM compared to OH_{obs}/OH_{mod} without HAM at different NO bins are presented in Supplementary Figure 13B. The decline rates for our seven warm-season campaigns, varying from 0-20% at different NO bins, show an increasing trend with the decrease in NO concentrations.

Supplementary Figure 13. (A) NO dependence on ratios of OH_{obs}/OH_{mod} under different sensitivity tests at different NO concentrations in the warm-season campaigns in China during daytime (around 08:00-15:00 local time). Sensitivity test 1: ‘w.o.HAM’ denotes that modeled OH concentrations are from the model without HAM. Sensitivity test 2: ‘w.HAM’ denotes that modeled OH concentrations are from the model with HAM. (B) Decline rates of OH_{obs}/OH_{mod} with HAM compared to OH_{obs}/OH_{mod} without HAM at different NO concentrations in the warm-season campaigns in China during daytime (around 08:00-15:00 local time).

We cited this paper and revised the description in original Lines 262-264: Under ultra-high NO concentrations, although carbonyl RO₂ autoxidation remains effective²⁵, the RAM cannot compete with the bimolecular reactions of RO₂ with NO anymore, so the effects of LIM and RAM on OH production become negligible (Supplementary Text 7).

Q7. The presentation of field data should be improved as they motivate much of the modeling work.
a. I would suggest presenting Figure 1A for all measurement sites separately in the SI without any NO-dependent binning.

Response: Thanks for your valuable suggestions. We have added the relationship between the ratio of observed to modeled OH concentrations and NO concentrations without binning in the revised Supplementary Fig. 3.

We revised the description of NO dependence on $\text{OH}_{\text{obs}}/\text{OH}_{\text{mod}}$ in the original Line 73: The OH underestimation becomes systematically more severe with the decreasing NO concentrations (Figure 1A and Supplementary Fig. 3).

Supplementary Figure 3. NO dependence on ratios of observed to modeled OH ($\text{OH}_{\text{obs}}/\text{OH}_{\text{mod}}$) in the seven campaigns in China.

b. The distributions of the observed NO concentrations and $\text{OH}_{\text{obs}}/\text{OH}_{\text{mod}}$ could be useful. These plots could help with justifying such binning shown in Fig. 1A – a description that remains missing.

Response: According to your helpful suggestions, we have added the distributions of the observed NO concentrations and $\text{OH}_{\text{obs}}/\text{OH}_{\text{mod}}$ in the revised Supplementary Information, as shown in the Response to Q7(a).

c. I strongly encourage you to show the range of variability in $\text{OH}_{\text{obs}}/\text{OH}_{\text{mod}}$ per each NO bin *e.g.*, with error bars.

Response: We have revised the Figure 1A, in which the range of variability in $\text{OH}_{\text{obs}}/\text{OH}_{\text{mod}}$ per each NO bin, as shown below.

Figure 1. The correlations between OH radicals and environmental parameters. (A) NO dependence on ratios of observed to modeled OH ($\text{OH}_{\text{obs}}/\text{OH}_{\text{mod}}$) during daytime (around 08:00-17:00 local time) covering both high NO and low NO conditions in the warm-season campaigns in China. The circles denote the median values, and the error bars denote the uncertainties.

d. It was also hard to notice at first glance that the x-axis in Fig. 1A is logarithmic. Could you perhaps add more tick marks to the axis to clarify this, or write it in the caption?

Response: We have revised Figure 1A in the revised manuscript, as shown in the Response to Q7c.

e. You mention in the SI that the Taizhou data points should be interpreted with caution because k_{OH} was not measured. Could it explain why the trend is not so clear for the lowest NO concentration for that given campaign?

Response: Thanks for your valuable suggestions. As you stated, the underestimation of k_{OH} would indeed cause the underestimation of missing OH sources. We further revised some related description in Line 109-112 in the original Supplementary Information: As more and more observed OVOCs are constrained in the model, coupled with the fact that numerous OVOCs have been simulated in the model, the simulated k_{OH} values are getting better and better matched with the measured values^{22, 23, 32}. Overall, due to the lack of k_{OH} measurement at the Taizhou site, the impact of missing k_{OH} on missing OH sources are inevitable, while this impact might be relatively small.

f. Fig. 1B shows one point per measurement site. Are these points campaign means? I suggest you add the error bars to these figures and provide information on where k_{OVOCs} , k_{AVOCs} , k_{BVOCs} , and k_{NO} are obtained from. Are all the data from low NO concentrations or just the OVOC panel?

Response: Thanks for your valuable suggestions which would improve the quality of our figures.

(1) The circle for each campaign in Figure 1B denotes the median value under low NO

concentrations during the entire campaign period (10:00-15:00 local time).

(2) We further added the error bars into the Figure 1B and Supplementary Fig. 5.

Figure 1. The correlations between OH radicals and environmental parameters. (A) NO dependence on ratios of observed to modeled OH (OH_{obs}/OH_{mod}) during daytime (around 08:00-17:00 local time) covering both high NO and low NO conditions in the warm-season campaigns in China. The circles denote the median values, and the error bars denote the uncertainties. These field campaigns include: Backgarden, Guangdong in summer 2006^{6, 33}; Yufa, Beijing in summer 2006²³; Wangdu, Hebei in summer 2014^{7, 22}; Heshan, Guangdong in autumn 2014¹¹; Taizhou, Jiangsu in summer 2018³⁴; Shenzhen, Guangdong in autumn 2018³⁵; Chengdu, Sichuan in summer 2019³⁶. (B) Correlations of missing OH sources with the matrix of OH reactivity, including ratios of anthropogenic VOCs (AVOCs) reactivity versus NO reactivity (k_{AVOCs}/k_{NO}), ratios of biogenic VOCs (BVOCs) reactivity versus NO reactivity (k_{BVOCs}/k_{NO}), and ratios of oxygenated VOCs (OVOCs) reactivity versus NO reactivity (k_{OVOCs}/k_{NO}) at low NO conditions (10:00-15:00 local time) in the seven warm-season campaigns in China. The circles denote the median values, and the error bars denote the uncertainties.

Supplementary Figure 5. Correlations of missing OH sources with the inverse of NO reactivity ($1/k_{NO}$), and ratios of VOCs reactivity versus NO reactivity (k_{VOCs}/k_{NO}) at low NO conditions (10:00-15:00 local time) in the seven warm-season campaigns in China. The circles denote the median values, and the error bars denote the uncertainties.

The detailed information on where k_{VOCs} , k_{OVOCs} , k_{AVOCs} , k_{BVOCs} , and k_{NO} are obtained from was further added in the last paragraph of Supplementary Text 1: In order to better explore the effect of

chemical species on radical chemistry, we further categorized k_{OH} into k_{NO} , k_{VOCs} , k_{AVOCs} , k_{BVOCs} , and k_{OVOCs} . k_{NO} denotes the NO reactivity, and k_{VOCs} denotes the difference between k_{OH} and inorganic reactivities (CO, NO_x, O₃, and SO₂). k_{AVOCs} represents the sum reactivities of alkanes, alkenes, and aromatics. k_{BVOCs} mainly represents the isoprene reactivity. k_{OVOCs} represents the k_{VOCs} minus the sum of k_{AVOCs} , k_{BVOCs} , and HCHO reactivity. In this case, k_{OVOCs} denotes all known and unknown OVOCs except HCHO. The removal of HCHO from k_{OVOCs} is mainly since the simple structure of HCHO makes the contribution of HCHO to radical chemistry clear.

(3) The data in Figure 1A is from the daytime (around 08:00-17:00 local time) and data in Figure 1B is only from low NO periods (10:00-15:00 local time), which is further explained in the above revised Figure 1 and Supplementary Fig. 5.

g. You state many places throughout the manuscript that they measure radical concentrations, but do not specify which radicals they measure.

Response: Thanks, we clarified it in “Methods: Radical observations” in the revised manuscript, as shown below: Both OH and HO₂ concentrations were measured in the seven field campaigns, and RO₂ measurements were carried out in the Wangdu and Heshan sites.

h. The laser-induced fluorescence system (LIF) based on fluorescence assay by gas expansion technique does not have a reference.

Response: We added several references related to the LIF system accordingly in “Methods: Radical observations” in the revised manuscript:

During these campaigns, radical concentrations are measured by the laser-induced fluorescence system (LIF) based on the fluorescence assay by gas expansion technique^{6,37}.

D. E. Heard, M. J. Pilling, Measurement of OH and HO₂ in the Troposphere. *Chemical Reviews* 103, 5163-5198 (2003).

K. D. Lu et al., Observation and modelling of OH and HO₂ concentrations in the Pearl River Delta 2006: a missing OH source in a VOC rich atmosphere. *Atmospheric Chemistry and Physics* 12, 1541-1569 (2012).

Q8. I suggest some grammar checks for the manuscript in addition to improving the clarity and precision of the text. At its current state, I do not believe the methodology is reproducible.

Response: Thanks for your valuable suggestions, we have done a careful and comprehensive grammar check for the manuscript accordingly, which largely enhanced the clarity and precision of the text.

With respect to the methodology reproductivity, since the field observation dataset used in this study have been published in peer reviewed journals^{6, 7, 11, 23, 33, 35, 38, 39}. Furthermore, the main analytical methods and research framework based our radical data have been well established and widely used in previous studies^{33, 40, 41}. We are aware of this misunderstanding may be caused by the grammar issues or missing details for the description of data, methods, and other supporting information. To

further ensure that methodology can be more easily following and reproducible by readers, we upload the related data and calculation to the figshare website proposed by the journal for open access, and added more detailed information for methodology and dataset in the revised manuscript, including:

- (1) More description on radical closure experiments have been added in Supplementary Text 2 (see below).
- (2) More information for OH recycling probability has been added in “Methods: Calculation of OH recycling probability” in the revised manuscript. The newly added information has been shown in the Response to Q2.
- (3) The method of OH yields from the VOC oxidation through HPC photolysis have been refined, as shown in Supplementary Text 5 (Response to Q1).
- (4) More detailed information for HAM scheme have been added in Supplementary Table 5 (see below).
- (5) More detailed information on the dataset for the HAM scheme, as mentioned by reviewer #1, including four tables and six figures, including Supplementary Table 8. Supplementary Tables 10-12. Supplementary Figure 6B. Supplementary Figures 14-18.

The update of radical closure experiment description.

As for the comparison between observed and simulated radical concentrations, a zero-dimensional chemical box model based on RACM2 was utilized in this study. The model was constrained to CO, NO, NO₂, CH₄, O₃, HONO, and VOCs concentrations and photolysis frequencies, water vapor, ambient temperature, and pressure as well. For RACM2, the measured detailed VOC species were lumped into several categories, which is introduced in previous studies^{6,7}. The H₂ and CH₄ mixing ratio was assumed to be 550 ppb and 1900 ppb, respectively. Other parameters were from the observations with 5-min time resolution, which represents the actual atmospheric environmental conditions in every day during campaigns. Given the significance of ambient temperature on the results of quantum chemical calculations, herein, it's worth mentioning that the temperature input in the model was the real ambient temperature observed in campaigns rather than a constant value. Additionally, a spin-up time with 2 days at the beginning of simulations was used to reach steady-state conditions for long-lived species. A fix dilution equivalent of 24-h lifetime was added in the model to represent the dry deposition. Based on the above configuration in the model, simulated radical concentrations would be obtained and then they will be compared to the observed concentrations to investigate whether the chemical mechanisms are correct, which is the most direct indication to identify whether the chemical mechanisms in the troposphere are complete or not.

Supplementary Table 5.

- ACO₃ and HKET denote acetyl peroxy radicals and hydroxy ketone, respectively. *j*MACR represents the photolysis rate constant of methacrolein (MACR).
- The rate constant of No. 1 is from our quantum chemical calculations.
- The rate constant of No. 2 is from Table 1 of Tan et al.¹¹.
- The reactions of Nos. 3-5 are from Wang et al.⁸ and the rate constants of both No. 3 and No. 4 are from Wang et al.⁸.
- As for No. 5, Wang et al.⁸ did not give the rate constant directly. However, as mentioned by Wang et al.⁸, the RO radicals could react with O₂ at effective rates of (0.5~5) × 10⁴ s⁻¹ with rate

coefficients of 10^{-15} ~ 10^{-14} cm³ molecules⁻¹ s⁻¹ according to the studies from Atkinson⁹. The radical in No. 5 is α -hydroxy radicals (R₁R₂COH), and Atkinson¹⁰ once reported the reactions of O₂ and the simplest α -hydroxy radicals (CH₂OH) with the $k(\text{CH}_2\text{OH}+\text{O}_2)$ is 9.0×10^{-12} cm³ molecules⁻¹ s⁻¹ at 298 K, which is larger than the rate constant of RO+O₂. The rate coefficients of CH₂OH+O₂ reactions reported in previous studies were summarized in IUPAC website (https://iupac-aeris.ipsl.fr/show_datasheets.php?category=Gas-phase+organics%3A+R_oxygen) and were all higher those that of RO+O₂ reactions. To obtain a conservative result, we chose the rate constant in No. 5 as 10⁴ s⁻¹.

- The photolysis rate of No. 6 is a conservative result from Liu et al.¹².

References:

1. Gonzalez C, Schlegel HB. An improved algorithm for reaction path following. *The Journal of Chemical Physics* **90**, 2154-2161 (1989).
2. Fukui K. THE PATH OF CHEMICAL-REACTIONS - THE IRC APPROACH. *Accounts of Chemical Research* **14**, 363-368 (1981).
3. Gonzalez C, Schlegel HB. REACTION-PATH FOLLOWING IN MASS-WEIGHTED INTERNAL COORDINATES. *Journal of Physical Chemistry* **94**, 5523-5527 (1990).
4. Eckart C. The penetration of a potential barrier by electrons. *Physical Review* **35**, 1303-1309 (1930).
5. Viegas LP, Jensen F. A computer-based solution to the oxidation kinetics of fluorinated and oxygenated volatile organic compounds. *Environmental Science-Atmospheres* **3**, 855-871 (2023).
6. Lu KD, et al. Observation and modelling of OH and HO₂ concentrations in the Pearl River Delta 2006: a missing OH source in a VOC rich atmosphere. *Atmospheric Chemistry and Physics* **12**, 1541-1569 (2012).
7. Tan Z, et al. Radical chemistry at a rural site (Wangdu) in the North China Plain: observation and model calculations of OH, HO₂ and RO₂ radicals. *Atmospheric Chemistry and Physics* **17**, 663-690 (2017).
8. Wang S-n, Wu R-r, Wang L-m. Role of Hydrogen Migrations in Carbonyl Peroxy Radicals in the Atmosphere. *Chinese Journal of Chemical Physics* **32**, 457-466 (2019).
9. Atkinson R. Atmospheric reactions of alkoxy and beta-hydroxyalkoxy radicals. *International Journal of Chemical Kinetics* **29**, 99-111 (1997).
10. Atkinson R. GAS-PHASE TROPOSPHERIC CHEMISTRY OF ORGANIC-COMPOUNDS. *Journal of Physical and Chemical Reference Data*, R1-& (1994).
11. Tan Z, et al. Experimental budgets of OH, HO₂, and RO₂ radicals and implications for ozone formation in the Pearl River Delta in China 2014. *Atmospheric Chemistry and Physics* **19**, 7129-7150 (2019).
12. Liu Z, Vinh Son N, Harvey J, Mueller J-F, Peeters J. The photolysis of alpha-hydroperoxycarbonyls. *Physical Chemistry Chemical Physics* **20**, 6970-6979 (2018).
13. Inomata S, et al. Laboratory measurements of emission factors of nonmethane volatile organic compounds from burning of Chinese crop residues. *Journal of Geophysical Research-Atmospheres* **120**, 5237-5252 (2015).
14. Zhang Y, et al. Emission inventory of carbonaceous pollutants from biomass burning in the Pearl River Delta Region, China. *Atmospheric Environment* **76**, 189-199 (2013).
15. Atamaleki A, et al. Emission of aldehydes from different cooking processes: a review study. *Air Quality Atmosphere and Health* **15**, 1183-1204 (2022).
16. Dominguez R, Gomez M, Fonseca S, Lorenzo JM. Effect of different cooking methods on lipid oxidation and formation of volatile compounds in foal meat. *Meat Science* **97**, 223-230 (2014).
17. Praske E, et al. Atmospheric autoxidation is increasingly important in urban and suburban North America.

- Proceedings of the National Academy of Sciences of the United States of America* **115**, 64-69 (2018).
18. Wang S, Wang L. The atmospheric oxidation of dimethyl, diethyl, and diisopropyl ethers. The role of the intramolecular hydrogen shift in peroxy radicals. *Physical Chemistry Chemical Physics* **18**, 7707-7714 (2016).
 19. Wang L. The Atmospheric Oxidation Mechanism of Benzyl Alcohol Initiated by OH Radicals: The Addition Channels. *Chemphyschem* **16**, 1542-1550 (2015).
 20. Wang L, Wang L. Atmospheric Oxidation Mechanism of Sabinene Initiated by the Hydroxyl Radicals. *Journal of Physical Chemistry A* **122**, 8783-8793 (2018).
 21. Bianchi F, *et al.* Highly Oxygenated Organic Molecules (HOM) from Gas-Phase Autoxidation Involving Peroxy Radicals: A Key Contributor to Atmospheric Aerosol. *Chemical Reviews* **119**, 3472-3509 (2019).
 22. Fuchs H, *et al.* OH reactivity at a rural site (Wangdu) in the North China Plain: contributions from OH reactants and experimental OH budget. *Atmospheric Chemistry and Physics* **17**, 645-661 (2017).
 23. Lu KD, *et al.* Missing OH source in a suburban environment near Beijing: observed and modelled OH and HO₂ concentrations in summer 2006. *Atmospheric Chemistry and Physics* **13**, 1057-1080 (2013).
 24. Lelieveld J, Peters W, Dentener FJ, Krol MC. Stability of tropospheric hydroxyl chemistry. *Journal of Geophysical Research-Atmospheres* **107**, (2002).
 25. Wang Z, *et al.* Efficient alkane oxidation under combustion engine and atmospheric conditions. *Communications Chemistry* **4**, 18 (2021).
 26. Barua S, Iyer S, Kumar A, Seal P, Rissanen M. An aldehyde as a rapid source of secondary aerosol precursors: Theoretical and experimental study of hexanal autoxidation. *EGUsphere* **2023**, 1-24 (2023).
 27. Iyer S, *et al.* Molecular rearrangement of bicyclic peroxy radicals is a key route to aerosol from aromatics. *Nature Communications* **14**, 4984 (2023).
 28. Wang S, Wu R, Berndt T, Ehn M, Wang L. Formation of Highly Oxidized Radicals and Multifunctional Products from the Atmospheric Oxidation of Alkylbenzenes. *Environmental Science & Technology* **51**, 8442-8449 (2017).
 29. Berndt T, *et al.* Hydroxyl radical-induced formation of highly oxidized organic compounds. *Nature Communications* **7**, (2016).
 30. Crounse JD, Nielsen LB, Jorgensen S, Kjaergaard HG, Wennberg PO. Autoxidation of Organic Compounds in the Atmosphere. *Journal of Physical Chemistry Letters* **4**, 3513-3520 (2013).
 31. McFiggans G, *et al.* Secondary organic aerosol reduced by mixture of atmospheric vapours. *Nature* **565**, 587-593 (2019).
 32. Lou S, *et al.* Atmospheric OH reactivities in the Pearl River Delta - China in summer 2006: measurement and model results. *Atmospheric Chemistry and Physics* **10**, 11243-11260 (2010).
 33. Hofzumahaus A, *et al.* Amplified Trace Gas Removal in the Troposphere. *Science* **324**, 1702-1704 (2009).
 34. Ma X, *et al.* OH and HO₂ radical chemistry at a suburban site during the EXPLORE-YRD campaign in 2018. *Atmospheric Chemistry and Physics* **22**, 7005-7028 (2022).
 35. Yang X, *et al.* Radical chemistry in the Pearl River Delta: observations and modeling of OH and HO₂ radicals in Shenzhen in 2018. *Atmospheric Chemistry and Physics* **22**, 12525-12542 (2022).
 36. Yang X, *et al.* Observations and modeling of OH and HO₂ radicals in Chengdu, China in summer 2019. *The Science of the total environment* **772**, 144829-144829 (2021).
 37. Heard DE, Pilling MJ. Measurement of OH and HO₂ in the Troposphere. *Chemical Reviews* **103**, 5163-5198 (2003).
 38. Yang XP, *et al.* Observations and modeling of OH and HO₂ radicals in Chengdu, China in summer 2019. *Science of the Total Environment* **772**, (2021).
 39. Ma XF, *et al.* OH and HO₂ radical chemistry at a suburban site during the EXPLORE-YRD campaign in 2018. *Atmospheric Chemistry and Physics* **22**, 7005-7028 (2022).

40. Lu K, *et al.* Exploring atmospheric free-radical chemistry in China: the self-cleansing capacity and the formation of secondary air pollution. *National Science Review* **6**, 579-594 (2019).
41. Rohrer F, *et al.* Maximum efficiency in the hydroxyl-radical-based self-cleansing of the troposphere. *Nature Geoscience* **7**, 559-563 (2014).
1. Gonzalez C, Schlegel HB. An improved algorithm for reaction path following. *The Journal of Chemical Physics* **90**, 2154-2161 (1989).
2. Fukui K. THE PATH OF CHEMICAL-REACTIONS - THE IRC APPROACH. *Accounts of Chemical Research* **14**, 363-368 (1981).
3. Gonzalez C, Schlegel HB. REACTION-PATH FOLLOWING IN MASS-WEIGHTED INTERNAL COORDINATES. *Journal of Physical Chemistry* **94**, 5523-5527 (1990).
4. Eckart C. The penetration of a potential barrier by electrons. *Physical Review* **35**, 1303-1309 (1930).
5. Viegas LP, Jensen F. A computer-based solution to the oxidation kinetics of fluorinated and oxygenated volatile organic compounds. *Environmental Science-Atmospheres* **3**, 855-871 (2023).
6. Lu KD, *et al.* Observation and modelling of OH and HO₂ concentrations in the Pearl River Delta 2006: a missing OH source in a VOC rich atmosphere. *Atmospheric Chemistry and Physics* **12**, 1541-1569 (2012).
7. Tan Z, *et al.* Radical chemistry at a rural site (Wangdu) in the North China Plain: observation and model calculations of OH, HO₂ and RO₂ radicals. *Atmospheric Chemistry and Physics* **17**, 663-690 (2017).
8. Wang S-n, Wu R-r, Wang L-m. Role of Hydrogen Migrations in Carbonyl Peroxy Radicals in the Atmosphere. *Chinese Journal of Chemical Physics* **32**, 457-466 (2019).
9. Atkinson R. Atmospheric reactions of alkoxy and beta-hydroxyalkoxy radicals. *International Journal of Chemical Kinetics* **29**, 99-111 (1997).
10. Atkinson R. GAS-PHASE TROPOSPHERIC CHEMISTRY OF ORGANIC-COMPOUNDS. *Journal of Physical and Chemical Reference Data*, R1-& (1994).
11. Tan Z, *et al.* Experimental budgets of OH, HO₂, and RO₂ radicals and implications for ozone formation in the Pearl River Delta in China 2014. *Atmospheric Chemistry and Physics* **19**, 7129-7150 (2019).
12. Liu Z, Vinh Son N, Harvey J, Mueller J-F, Peeters J. The photolysis of alpha-hydroperoxycarbonyls. *Physical Chemistry Chemical Physics* **20**, 6970-6979 (2018).
13. Inomata S, *et al.* Laboratory measurements of emission factors of nonmethane volatile organic compounds from burning of Chinese crop residues. *Journal of Geophysical Research-Atmospheres* **120**, 5237-5252 (2015).
14. Zhang Y, *et al.* Emission inventory of carbonaceous pollutants from biomass burning in the Pearl River Delta Region, China. *Atmospheric Environment* **76**, 189-199 (2013).
15. Atamaleki A, *et al.* Emission of aldehydes from different cooking processes: a review study. *Air Quality Atmosphere and Health* **15**, 1183-1204 (2022).
16. Dominguez R, Gomez M, Fonseca S, Lorenzo JM. Effect of different cooking methods on lipid oxidation and formation of volatile compounds in foal meat. *Meat Science* **97**, 223-230 (2014).
17. Praske E, *et al.* Atmospheric autoxidation is increasingly important in urban and suburban North America. *Proceedings of the National Academy of Sciences of the United States of America* **115**, 64-69 (2018).
18. Wang S, Wang L. The atmospheric oxidation of dimethyl, diethyl, and diisopropyl ethers. The role of the intramolecular hydrogen shift in peroxy radicals. *Physical Chemistry Chemical Physics* **18**, 7707-7714 (2016).
19. Wang L. The Atmospheric Oxidation Mechanism of Benzyl Alcohol Initiated by OH Radicals: The Addition Channels. *Chemphyschem* **16**, 1542-1550 (2015).
20. Wang L, Wang L. Atmospheric Oxidation Mechanism of Sabinene Initiated by the Hydroxyl Radicals. *Journal of Physical Chemistry A* **122**, 8783-8793 (2018).
21. Bianchi F, *et al.* Highly Oxygenated Organic Molecules (HOM) from Gas-Phase Autoxidation Involving Peroxy

- Radicals: A Key Contributor to Atmospheric Aerosol. *Chemical Reviews* **119**, 3472-3509 (2019).
22. Fuchs H, *et al.* OH reactivity at a rural site (Wangdu) in the North China Plain: contributions from OH reactants and experimental OH budget. *Atmospheric Chemistry and Physics* **17**, 645-661 (2017).
 23. Lu KD, *et al.* Missing OH source in a suburban environment near Beijing: observed and modelled OH and HO₂ concentrations in summer 2006. *Atmospheric Chemistry and Physics* **13**, 1057-1080 (2013).
 24. Lelieveld J, Peters W, Dentener FJ, Krol MC. Stability of tropospheric hydroxyl chemistry. *Journal of Geophysical Research-Atmospheres* **107**, (2002).
 25. Wang Z, *et al.* Efficient alkane oxidation under combustion engine and atmospheric conditions. *Communications Chemistry* **4**, 18 (2021).
 26. Barua S, Iyer S, Kumar A, Seal P, Rissanen M. An aldehyde as a rapid source of secondary aerosol precursors: Theoretical and experimental study of hexanal autoxidation. *EGU sphere* **2023**, 1-24 (2023).
 27. Iyer S, *et al.* Molecular rearrangement of bicyclic peroxy radicals is a key route to aerosol from aromatics. *Nature Communications* **14**, 4984 (2023).
 28. Wang S, Wu R, Berndt T, Ehn M, Wang L. Formation of Highly Oxidized Radicals and Multifunctional Products from the Atmospheric Oxidation of Alkylbenzenes. *Environmental Science & Technology* **51**, 8442-8449 (2017).
 29. Berndt T, *et al.* Hydroxyl radical-induced formation of highly oxidized organic compounds. *Nature Communications* **7**, (2016).
 30. Crounse JD, Nielsen LB, Jorgensen S, Kjaergaard HG, Wennberg PO. Autoxidation of Organic Compounds in the Atmosphere. *Journal of Physical Chemistry Letters* **4**, 3513-3520 (2013).
 31. McFiggans G, *et al.* Secondary organic aerosol reduced by mixture of atmospheric vapours. *Nature* **565**, 587-593 (2019).
 32. Lou S, *et al.* Atmospheric OH reactivities in the Pearl River Delta - China in summer 2006: measurement and model results. *Atmospheric Chemistry and Physics* **10**, 11243-11260 (2010).
 33. Hofzumahaus A, *et al.* Amplified Trace Gas Removal in the Troposphere. *Science* **324**, 1702-1704 (2009).
 34. Ma X, *et al.* OH and HO₂ radical chemistry at a suburban site during the EXPLORE-YRD campaign in 2018. *Atmospheric Chemistry and Physics* **22**, 7005-7028 (2022).
 35. Yang X, *et al.* Radical chemistry in the Pearl River Delta: observations and modeling of OH and HO₂ radicals in Shenzhen in 2018. *Atmospheric Chemistry and Physics* **22**, 12525-12542 (2022).
 36. Yang X, *et al.* Observations and modeling of OH and HO₂ radicals in Chengdu, China in summer 2019. *The Science of the total environment* **772**, 144829-144829 (2021).
 37. Heard DE, Pilling MJ. Measurement of OH and HO₂ in the Troposphere. *Chemical Reviews* **103**, 5163-5198 (2003).
 38. Yang XP, *et al.* Observations and modeling of OH and HO₂ radicals in Chengdu, China in summer 2019. *Science of the Total Environment* **772**, (2021).
 39. Ma XF, *et al.* OH and HO₂ radical chemistry at a suburban site during the EXPLORE-YRD campaign in 2018. *Atmospheric Chemistry and Physics* **22**, 7005-7028 (2022).
 40. Lu K, *et al.* Exploring atmospheric free-radical chemistry in China: the self-cleansing capacity and the formation of secondary air pollution. *National Science Review* **6**, 579-594 (2019).
 41. Rohrer F, *et al.* Maximum efficiency in the hydroxyl-radical-based self-cleansing of the troposphere. *Nature Geoscience* **7**, 559-563 (2014).

Reviewer #1 (Remarks to the Author):

There is no doubt that the authors exerted extra effort to respond to all comments and questions made by the reviewers. With respect to my comments/questions, I am satisfied (in general) to the new information provided by the authors.

I only have minor corrections and suggestions:

1 - I believe it is more correct to say "rotatable bonds" instead of "rotation bonds", so I would change this in the Supplementary Information.

2 - The authors state that they obtain 63 distinguishable conformers for the CH3CH2CH2CH2CH2C(O)O2 reactant. If I take this molecule's Cartesian coordinates from the Supplementary Information and use them on Confab (with default settings), I obtain 1728 conformers:

```
**Starting Confab 1.1.0
```

```
**To support, cite Journal of Cheminformatics, 2011, 3, 8.
```

```
..Input format = xyz
```

```
..Output format = sy2
```

```
..RMSD cutoff = 0.05
```

```
..Energy cutoff = 500
```

```
..Conformer cutoff = 1000000
```

```
..Write input conformation? False
```

```
..Verbose? False
```

```
**Molecule 1
```

```
..title = reactant.xyz
```

```
..number of rotatable bonds = 5
```

```
..tot conformations = 1944
```

```
..tot confs tested = 1944
```

```
..below energy threshold = 1728
```

```
..generated 1728 conformers
```

And this number can rise up to 1857, if one of those 1728 conformers is used again as "seed" to a new calculation in Confab. Of course, this is based on force fields and after a better calculation, the number of found conformers will most likely decrease. My question is: how did the authors arrive at the number 63? What calculations did they perform? Did they use some code (like Confab, or similar) to get a first set of conformers, and then they refined this number with a better electronic structure calculation? The authors should clarify this.

3 - The two-time used expression "producing conformers" is strange. Please remove this or change the way those sentences are written.

4 - In their rebuttal file, the authors reply to one of my initial questions as:

"Q2. Do the five shown TSs all connect to that one specific reactant conformer as shown in Figure S6?"

Response: Yes, we had done the IRC calculation to verify the TSs correctly connect to the specific reactant conformer."

But the answer should clearly be "No.". By looking at the energies and drawings of the reactant conformers in Supplementary Figures 14 to 18, the reactant conformers connected to each of the TSs are different. Please include in those figures what reactant conformer does the first IRC point correspond to. Also, is the IRC path in Supplementary Figure 14 well converged in the reactants direction? It looks like it still has some slope there.

5 - Please provide Cartesian coordinates for ALL (not just the lowest energy ones) calculated reactant and TS conformers, as the data in Supplementary Table 8 should be reproducible. These

can be in the form of .xyz files inside a zip.

6 - Please provide the value for ALL (not just the lowest energy ones) imaginary frequencies of the TS conformers.

Reviewer #2 (Remarks to the Author):

The authors have clarified all the comments and improved the quality of the manuscript. I recommend it to be accepted.

Reviewer #3 (Remarks to the Author):

The paper demonstrates that aldehydes can act as a major source of OH radicals in low NO environment. This novel hypothesis provides an explanation why models struggle reproducing OH concentration under low NO concentrations. The work has high significance for both climate and air quality as OH radicals greatly influence e.g. atmospheric methane concentrations and oxidation capacity in general.

The authors have nicely addressed all my previous comments during the first review round. The manuscript looks better, and I believe the methodology description has greatly improved towards being reproducible!

Some comments/suggestions below for further improvement:

1. Thank you for the HO₂ vs NO plots (SI Fig. 8). As the OH yields from aldehydes are currently estimated using the inverse modeling techniques by selecting yields that guarantee a good agreement between modeled and measured OH, I find it interesting to see how other radicals are affected by the reactive aldehyde chemistry, and whether any improvements can be seen e.g., in HO₂ concentrations between modeled and measured having the new chemistry is included in the model.

- a. In SI Fig. 8, I can see that the green markers represent Base+RAM. Does this mean that LIM1 was not involved (sorry if I missed this)?
- b. Assuming LIM1 is not included, how would including it change the picture?
- c. Can you really say that the HO₂ improves (to some extent) when RAM is involved? I find it hard to say from this figure. Can you quantify the improvement with a scatter plot + regression analysis (HO₂ obs vs HO₂ meas)?
- d. Can these plots be made also for RO₂?

2. While the conclusions of the study will not be affected, I believe for that for the sake of credibility of the statistical analyses, Figure 1 should still be further improved.

- a. Error bars are currently stated to describe uncertainty in the data. What do you mean by uncertainty? Does this mean measurement uncertainty involving some estimated measurement error or do you mean variability in the data?
- b. My suggestion is to show the data variability. Because the markers are median values, I would draw the error bars between 25th and 75th percentiles within the data. I would say that utilizing standard deviation (which is maybe what you show?) makes sense only if your data is normally distributed, which I assume is not the case.
- c. I would also add error bars in the x direction to show the width of the NO bins.
- d. Maybe the error bars could be same color as the markers?
- e. Could you add confidence intervals to Fig. 1B using the data variability?

Technical:

- L70 "It is demonstrated" → "this study demonstrates" or "It is demonstrated" + reference.
- L208 define "high carbon numbers"
- L411 define "ALD"

Response to Reviewer #1:

General comments:

There is no doubt that the authors exerted extra effort to respond to all comments and questions made by the reviewers. With respect to my comments/questions, I am satisfied (in general) to the new information provided by the authors. I only have minor corrections and suggestions.

Response: We appreciate the reviewer's positive comments about the revised manuscript. Below are our responses to the specific comments and revisions.

Specific comments:

Q1. I believe it is more correct to say "rotatable bonds" instead of "rotation bonds", so I would change this in the Supplementary Information.

Response: Thanks for your suggestions. We have corrected it.

Q2. The authors state that they obtain 63 distinguishable conformers for the $\text{CH}_3\text{CH}_2\text{CH}_2\text{CH}_2\text{CH}_2\text{C}(\text{O})\text{O}_2$ reactant. If I take this molecule's Cartesian coordinates from the Supplementary Information and use them on Confab (with default settings), I obtain 1728 conformers:

```
**Starting Confab 1.1.0
```

```
**To support, cite Journal of Cheminformatics, 2011, 3, 8.
```

```
..Input format = xyz
```

```
..Output format = sy2
```

```
..RMSD cutoff = 0.05
```

```
..Energy cutoff = 500
```

```
..Conformer cutoff = 1000000
```

```
..Write input conformation? False
```

```
..Verbose? False
```

```
**Molecule 1
```

```
..title = reactant.xyz
```

```
..number of rotatable bonds = 5
```

```
..tot conformations = 1944
```

```
..tot confs tested = 1944
```

```
..below energy threshold = 1728
```

```
..generated 1728 conformers
```

And this number can rise up to 1857, if one of those 1728 conformers is used again as "seed" to a new calculation in Confab. Of course, this is based on force fields and after a better calculation, the number of found conformers will

most likely decrease. My question is: how did the authors arrive at the number 63? What calculations did they perform? Did they use some code (like Confab, or similar) to get a first set of conformers, and then they refined this number with a better electronic structure calculation? The authors should clarify this.

Response: In our calculations, the initial guess conformers for $\text{CH}_3\text{CH}_2\text{CH}_2\text{CH}_2\text{CH}_2\text{C}(\text{O})\text{O}_2$ is 237 generated by using MSTor 2017 program^{1,2}. These initial conformers were firstly optimized by B3LYP³/6-31G(d,p) method. As a result, there are only 70 distinguishable conformers from B3LYP/6-31G(d,p) calculated results. Furthermore, the 70 distinguishable conformers were reoptimized by using M06-2X⁴/MG3S⁵ method to lead to the formation of 63 distinguishable conformers. The input of the $\text{CH}_3\text{CH}_2\text{CH}_2\text{CH}_2\text{CH}_2\text{C}(\text{O})\text{O}_2$ reactant for obtaining conformer is provided as follows.

```

20 5
C      1.93363100    0.50842200    0.00000000
O      2.00225900    1.68737600    0.00000000
C      0.71317200   -0.35664200    0.00000000
H      0.76473700   -1.01783300   -0.86756900
H      0.76473700   -1.01783300    0.86756900
O      3.19051100   -0.17040200    0.00000000
O      3.09777900   -1.46729200    0.00000000
C     -0.55795400    0.47962900    0.00000000
H     -0.55920800    1.13584000   -0.87246400
H     -0.55920800    1.13583900    0.87246400
C     -1.80898200   -0.38994200    0.00000000
H     -1.79837000   -1.04683700    0.87547000
H     -1.79837000   -1.04683600   -0.87547100
C     -3.09551400    0.42760400    0.00000000
H     -3.10324600    1.08285900   -0.87457700
H     -3.10324600    1.08285900    0.87457600
C     -4.34260300   -0.44800400    0.00000000
H     -4.36525300   -1.09141700    0.88052500
H     -5.25221400    0.15092200    0.00000000
H     -4.36525300   -1.09141700   -0.88052500

```

#torsion 1 definition

```

1 6
2
6 7
3
0.0 120 240

```

#torsion 2 definition

```

1 3
4
1 2 6 7
3
0.0 120 240

```

#torsion 3 definition

```

3 8
7
1 2 3 4 5 6 7
3
0.0 120 240

```

#torsion 4 definition

```

8 11
10

```

```
1 2 3 4 5 6 7 8 9 10
3
0.0 120 240
#torsion 5 definition
11 14
13
1 2 3 4 5 6 7 8 9 10 11 12 13
3
0.0 120 240
%mem=32GB
%nprocshared=16
# opt=(Calcfc,Tight,maxcycle=200,maxstep=14)   freq=noraman   ub3lyp/6-31G(d,p)   Int(grid=99974)
scf=(Conver=10)   int=Acc2E=10
0 2
```

Other input files were provided in a separate file named xyz.zip file.

Q3. The two-time used expression "producing conformers" is strange. Please remove this or change the way those sentences are written.

Response: Thank you for your suggestion! We have rewritten the corresponding sentences in the revised Supplementary Information.

Q4. In their rebuttal file, the authors reply to one of my initial questions as:

"Q2. Do the five shown TSs all connect to that one specific reactant conformer as shown in Figure S6?

Response: Yes, we had done the IRC calculation to verify the TSs correctly connect to the specific reactant conformer."

But the answer should clearly be "No.". By looking at the energies and drawings of the reactant conformers in Supplementary Figures 14 to 18, the reactant conformers connected to each of the TSs are different. Please include in those figures what reactant conformer does the first IRC point correspond to. Also, is the IRC path in Supplementary Figure 14 well converged in the reactants direction? It looks like it still has some slope there.

Response: We agree with your comments. We have added some comments in Supplementary Information: We also did intrinsic reaction coordinate (IRC) calculations to show that the transition state connects with the corresponding reactant and intermediate product conformer as listed in Supplementary Figures 14-18. However, in Supplementary Figure 6A, the reactant and intermediate product are both the lowest conformers of the reactants and intermediates. The lowest conformers can facily convert to the specific conformer along internal rotation, which is connected with the corresponding transition state.

Q5. Please provide Cartesian coordinates for ALL (not just the lowest energy ones) calculated reactant and TS conformers, as the data in Supplementary Table 8 should be reproducible. These can be in the form of .xyz files inside a zip.

Response: We have provided Cartesian coordinates for all calculated reactant and transition state conformers in a separate file with named as xyz.zip.

Q6. Please provide the value for ALL (not just the lowest energy ones) imaginary frequencies of the TS conformers.

Response: The value for all imaginary frequencies of the TS conformers was provided in an Excel file named imaginary frequencies in the xyz.zip file.

References:

1. Zheng J, Mielke SL, Clarkson KL, Truhlar DG. *MSTor*: A program for calculating partition functions, free energies, enthalpies, entropies, and heat capacities of complex molecules including torsional anharmonicity. *Computer Physics Communications* **183**, 1803-1812 (2012).
2. Zheng J, Mielke SL, Bao JL, Meana-Pafeda R, Clarkson KL, Truhlar DG. *MSTor* computer program, version 2017-B. *University of Minnesota, Minneapolis, MN*, (2017).
3. Stephens PJ, Devlin FJ, Chabalowski CF, Frisch MJ. Ab Initio Calculation of Vibrational Absorption and Circular Dichroism Spectra Using Density Functional Force Fields. *The Journal of Physical Chemistry* **98**, 11623-11627 (1994).
4. Zhao Y, Truhlar DG. The M06 suite of density functionals for main group thermochemistry, thermochemical kinetics, noncovalent interactions, excited states, and transition elements: two new functionals and systematic testing of four M06-class functionals and 12 other functionals. *Theoretical Chemistry Accounts* **120**, 215-241 (2008).
5. Lynch BJ, Zhao Y, Truhlar DG. Effectiveness of diffuse basis functions for calculating relative energies by density functional theory. *Journal of Physical Chemistry A* **107**, 1384-1388 (2003).

Response to Reviewer #3:

General comments:

The paper demonstrates that aldehydes can act as a major source of OH radicals in low NO environment. This novel hypothesis provides an explanation why models struggle reproducing OH concentration under low NO concentrations. The work has high significance for both climate and air quality as OH radicals greatly influence e.g. atmospheric methane concentrations and oxidation capacity in general.

The authors have nicely addressed all my previous comments during the first review round. The manuscript looks better, and I believe the methodology description has greatly improved towards being reproducible!

Some comments/suggestions below for further improvement.

Response: Thank you very much for recognizing our paper and your constructive comments, which are critical to improving our manuscript. Below are our responses to the specific comments and revision.

Specific comments:

Q1. Thank you for the HO₂ vs NO plots (SI Fig. 8). As the OH yields from aldehydes are currently estimated using the inverse modeling techniques by selecting yields that guarantee a good agreement between modeled and measured OH, I find it interesting to see how other radicals are affected by the reactive aldehyde chemistry, and whether any improvements can be seen e.g., in HO₂ concentrations between modeled and measured having the new chemistry is included in the model.

a. In SI Fig. 8, I can see that the green markers represent Base+RAM. Does this mean that LIM1 was not involved (sorry if I missed this)?

Response: We have coupled the LIM1 into the model in the “Base” scene in the previous manuscript. To further clarify it, we changed “Base” to “Base + LIM1” and changed “Base + RAM” to “Base + LIM1 + RAM” in the revised Supplementary Figure 8. In addition, according to the suggestions in Q1 (c), compared to the NO dependence, we think the scatter plot and regression analysis might be a better presentation of the effect of RAM on HO₂ radicals. Thus, we replaced it with the scatter plot as shown in detail in Q1 (c).

b. Assuming LIM1 is not included, how would including it change the picture?

Response: We have evaluated the effect of the LIM1 on HO₂ simulations. The evaluation results indicated that the LIM1 elevate the simulated HO₂ concentrations with a minor contribution by 1%-13% among our seven warm-season campaigns with an average of 6%. Thus, we based the “Base + LIM1” scenario to assess the influence of RAM on radicals further.

c. Can you really say that the HO₂ improves (to some extent) when RAM is involves? I find it hard to say from this figure. Can you quantify the improvement with a scatter plot + regression analysis (HO₂ obs vs HO₂ meas)?

Response: Thanks, we conducted the scatter plot and regression analysis for the observed and simulated HO₂ concentrations accordingly. As shown in the following figure, the slopes of the linear fitting curve from the “Base + LIM1 + RAM” scenario to observation are all closer to 1 than the slopes of the linear fitting curve from the “Base + LIM1” scenario, indicated a better agreement of the modeled and observed HO₂ concentrations. Considering that the original Supplementary Figure 8 cannot show a clear message about the improvement, we replaced the original Supplementary Figure 8 with the following one.

Supplementary Figure 8. Correlations between the observed and modeled HO₂ concentrations in our seven warm-season campaigns. The modeling results include two scenes, one is the model incorporated with LIM1 (blue color) and the other is the model incorporated with LIM1 and RAM together (red color). Only daytime values and NO concentration above the detection limit of the instrument were chosen.

d. Can these plots be made also for RO₂?

Response: Thanks for the suggestion. The comparison to RO₂ cannot be achieved in the current stage for the following reasons: (1) the detection of RO₂ by laser-induced fluorescence (LIF) technique is the sum of RO₂ radicals¹, making it difficult to assess the newly proposed chemical mechanisms that require the measurement of specific RO₂ groups or species. (2) The RO₂ measurement based on the LIF technique is an indirect method, in which RO₂ radicals are transformed into HO₂ radicals which are then converted into OH radicals¹, at the current stage, the physical meanings of the measured total RO₂ concentrations are yet to be discussed further. (3) Only few RO₂ measurements have been conducted including Wangdu², Heshan³, and Chengdu⁴ sites among our seven warm-season campaigns, making it difficult to get systematic results that are representative of the whole fourteen campaigns. Therefore, these plots unfortunately cannot be made for RO₂ at current stage but will be a very valuable direction to explore in the near future.

Q2. While the conclusions of the study will not be affected, I believe for that for the sake of credibility of the statistical analyses, Figure 1 should still be further improved.

a. Error bars are currently stated to describe uncertainty in the data. What do you mean by uncertainty? Does this mean measurement uncertainty involving some estimated measurement error or do you mean variability in the data?

Response: In the old version, Figure 1 shows an estimated uncertainty from measurements and simulations in the parameter calculation process.

b. My suggestion is to show the data variability. Because the markers are median values, I would draw the error bars between 25th and 75th percentiles within the data. I would say that utilizing standard deviation (which is maybe what you show?) makes sense only if your data is normally distributed, which I assume is not the case.

Response: Thanks for your valuable suggestions. We have changed the representation of the error bars in Figure 1 from uncertainty to variability (25th to 75th percentiles), and we have also taken Q1 (c, d, e) into account in the revision of Figure 1. Accordingly, we have revised the Supplementary Figure 5.

Figure 1. The correlations between OH radicals and environmental parameters. (A) NO dependence on ratios of observed to modeled OH (OH_{obs}/OH_{mod}) during daytime (around 08:00-17:00 local time) covering both high NO and low NO conditions in the warm-season campaigns in China. These field campaigns include: Backgarden,

Guangdong in summer 2006^{5, 6}; Yufa, Beijing in summer 2006⁷; Wangdu, Hebei in summer 2014^{2, 8}; Heshan, Guangdong in autumn 2014³; Taizhou, Jiangsu in summer 2018⁹; Shenzhen, Guangdong in autumn 2018¹⁰; Chengdu, Sichuan in summer 2019¹¹. The circles denote the median values, and the error bars denote the 25th to 75th percentiles. (B) Correlations of missing OH sources with the matrix of OH reactivity, including ratios of anthropogenic VOCs (AVOCs) reactivity versus NO reactivity (k_{AVOCs}/k_{NO}), ratios of biogenic VOCs (BVOCs) reactivity versus NO reactivity (k_{BVOCs}/k_{NO}), and ratios of oxygenated VOCs (OVOCs) reactivity versus NO reactivity (k_{OVOCs}/k_{NO}) at low NO conditions (10:00-15:00 local time) in the seven warm-season campaigns in China. The circles denote the median values, and the error bars denote the 25th to 75th percentiles. The black dotted line indicates the linear fitting curve. The pink shading shows the 95% confidence intervals.

Supplementary Figure 5. Correlations of missing OH sources with the inverse of NO reactivity ($1/k_{NO}$), and ratios of VOCs reactivity versus NO reactivity (k_{VOCs}/k_{NO}) at low NO conditions (10:00-15:00 local time) in the seven warm-season campaigns in China. The circles denote the median values, and the error bars denote the 25th to 75th percentiles. The black dotted line indicates the linear fitting curve. The pink shading shows the 95% confidence intervals.

c. I would also add error bars in the x direction to show the width of the NO bins.

Response: Revised accordingly, as shown in the Response to Q1 (b).

d. Maybe the error bars could be same color as the markers?

Response: Revised accordingly, as shown in the Response to Q1 (b).

e. Could you add confidence intervals to Fig. 1B using the data variability?

Response: Added 95% confidence intervals to Fig. 1B, as shown in the Response to Q1 (b).

Technical:

- L70 “It is demonstrated” → “this study demonstrates” or “It is demonstrated” + reference.

Response: Revised accordingly.

It is demonstrated that more detailed mechanisms are urgently needed to narrow the gaps of OH radical chemical mechanisms in the troposphere^{5, 12, 13, 14, 15}.

- L208 define “high carbon numbers”

Response: We changed: Aldehydes from primary emissions, especially aldehydes with high carbon numbers ($\geq C8/C9$) from biomass burning or cooking activities.

- L411 define “ALD”

Response: We have added the definition for “ALD”: ALD (propanal, butanal, and larger R-groups aldehydes).

References

1. Fuchs H, Holland F, Hofzumahaus A. Measurement of tropospheric HO_2 and HO_2 radicals by a laser-induced fluorescence instrument. *Review of Scientific Instruments* **79**, (2008).
2. Tan Z, *et al.* Radical chemistry at a rural site (Wangdu) in the North China Plain: observation and model calculations of OH, HO₂ and RO₂ radicals. *Atmospheric Chemistry and Physics* **17**, 663-690 (2017).
3. Tan Z, *et al.* Experimental budgets of OH, HO₂, and RO₂ radicals and implications for ozone formation in the Pearl River Delta in China 2014. *Atmospheric Chemistry and Physics* **19**, 7129-7150 (2019).
4. Li S, Lu K, Ma X, Yang X, Chen S, Zhang Y. Field measurement of the organic peroxy radicals by the low-pressure reactor plus laser-induced fluorescence spectroscopy. *Chinese Chemical Letters* **31**, 2799-2802 (2020).
5. Hofzumahaus A, *et al.* Amplified Trace Gas Removal in the Troposphere. *Science* **324**, 1702-1704 (2009).
6. Lu KD, *et al.* Observation and modelling of OH and HO₂ concentrations in the Pearl River Delta 2006: a missing OH source in a VOC rich atmosphere. *Atmospheric Chemistry and Physics* **12**, 1541-1569 (2012).
7. Lu KD, *et al.* Missing OH source in a suburban environment near Beijing: observed and modelled OH and HO₂ concentrations in summer 2006. *Atmospheric Chemistry and Physics* **13**, 1057-1080 (2013).
8. Fuchs H, *et al.* OH reactivity at a rural site (Wangdu) in the North China Plain: contributions from OH reactants and experimental OH budget. *Atmospheric Chemistry and Physics* **17**, 645-661 (2017).
9. Ma X, *et al.* OH and HO₂ radical chemistry at a suburban site during the EXPLORE-YRD campaign in 2018. *Atmospheric Chemistry and Physics* **22**, 7005-7028 (2022).
10. Yang X, *et al.* Radical chemistry in the Pearl River Delta: observations and modeling of OH and HO₂ radicals in Shenzhen in 2018. *Atmospheric Chemistry and Physics* **22**, 12525-12542 (2022).
11. Yang X, *et al.* Observations and modeling of OH and HO₂ radicals in Chengdu, China in summer 2019. *The Science of the total environment* **772**, 144829-144829 (2021).
12. Peeters J, Muller J-F, Stavrou T, Vinh Son N. Hydroxyl Radical Recycling in Isoprene Oxidation Driven by Hydrogen Bonding and Hydrogen Tunneling: The Upgraded LIM1 Mechanism. *Journal of Physical Chemistry A* **118**, 8625-8643 (2014).
13. Peeters J, Nguyen TL, Vereecken L. HO_x radical regeneration in the oxidation of isoprene. *Physical Chemistry Chemical Physics* **11**, 5935-5939 (2009).
14. Thornton JA, *et al.* Ozone production rates as a function of NO_x abundances and HO_x production rates in the Nashville urban plume. *Journal of Geophysical Research-Atmospheres* **107**, (2002).
15. Lelieveld J, *et al.* Atmospheric oxidation capacity sustained by a tropical forest. *Nature* **452**, 737-740 (2008).

Reviewer #3 (Remarks to the Author):

I am very happy with the current state of the paper. I have no further comments.